EMBO
Molecular Medicine

# Metabolic signatures for gastric cancer diagnosis and mechanistic insights: a multicenter study

Juan Zhu[1,2,3,10], Yida Huang (iD) [4,5,10], Xue Li[1,2,3,10], Bin Liu[6,10], Li Yuan[2,3,7], Le Wang[1], Kun Qian (iD) [4,5], Yingying Mao (iD) [6], Yongjie Xu (iD) [8✉], Lingbin Du (iD) [1,9✉] & Xiangdong Cheng (iD) [1,2,3,7✉]

## Abstract

Early detection of gastric cancer (GC) is critical for improving prognosis, yet conventional biomarkers lack sensitivity and specificity, necessitating non-invasive, high-performance diagnostic tools. This study integrated untargeted metabolomics and machine learning to develop a plasma metabolite panel for GC diagnosis and mechanistic insights. Plasma and tissue samples from two cohorts ($n = 597$) were analyzed using ultra-performance liquid chromatography–mass spectrometry (UPLC-MS). A six-metabolite panel was identified and validated, demonstrating excellent diagnostic performance (area under the curve: 0.947–0.982 in discovery; 0.920–0.951 in validation) and superior sensitivity (0.900–0.940) compared to conventional markers (0.020–0.240). Isovalerylcarnitine (C5), a key component, was consistently downregulated in both plasma and tissue samples. Mendelian randomization supported a causal relationship between isovalerylcarnitine (C5) and GC risk. Proteomic analyses revealed inverse correlations between C5 and cadherin/MMP family proteins. Functional assays confirmed that isovalerylcarnitine (C5) inhibited GC cell migration and invasion via calpain-mediated cleavage of VE-cadherin and MMP2. This study identifies a robust diagnostic metabolite panel for GC detection and highlights a novel mechanistic role of isovalerylcarnitine (C5) in GC progression, supporting its utility as both a biomarker and therapeutic target.

**Keywords** Gastric Cancer; Metabolomics; Biomarker; Early Detection; Isovalerylcarnitine (C5)
**Subject Categories** Biomarkers; Cancer; Digestive System

## Introduction

According to the latest GLOBOCAN 2022 data, gastric cancer (GC) remains a global health burden (Chen et al, 2024b; Thrift et al, 2023; Morgan et al, 2022), ranking fifth in both incidence and mortality worldwide (Bray et al, 2024). Notably, China accounts for 37.0% of new GC cases and 39.4% of GC deaths (Han et al, 2024). Early detection is crucial for improving treatment outcomes, reducing mortality (Chen et al, 2020) and increasing the 5-year survival rate (Zeng et al, 2018). However, current GC diagnosis methods mainly rely on gastroscopy, facing significant limitations, including low participation rates (as low as 33.5%) due to its invasive nature and dependence on physician expertise, thereby hindering large-scale screening efforts (Chen et al, 2020). Thus, there is a pressing demand for noninvasive, cost-effective, and robust diagnostic tools, such as biomarker-based blood tests, to facilitate early GC detection (Ma et al, 2023; Joshi and Badgwell, 2021).

The development of effective blood-based diagnostic tests hinges on the identification of reliable biomarkers. Traditional blood protein biomarkers (e.g., CEA, CA199, CA724) exhibit limited clinical utility in GC detection due to poor sensitivity and specificity (Xu et al, 2021). Recent research has explored a range of noninvasive biomarkers, from cellular (e.g., circulating tumor cells and exosomes) to molecular (e.g., proteins and metabolites) levels (Chen et al, 2022; Maron et al, 2019; So et al, 2021; Shinozuka et al, 2023; Huang et al, 2021). Among these, molecular biomarkers, particularly metabolites, have gained prominence due to their stability and superior diagnostic performance (El-Deiry et al, 2019). In contrast to upstream molecular biomarkers (e.g., nucleic acids and proteins), downstream metabolic biomarkers directly reflect the body's phenotypic states, thereby serving as promising candidates for high-performance diagnostic tests (Bar et al, 2020; Buergel et al, 2022). Furthermore, metabolites, as the final products of gene and protein activities, amplify subtle gene and protein changes, serving as dynamic and sensitive indicators of phenotypic changes (Buergel et al, 2022; Schmidt et al, 2021). Metabolomics is emerging as a promising approach for cancer prevention, diagnosis,

[1]Department of Cancer Prevention, Zhejiang Cancer Hospital, Hangzhou Institute of Medicine (HIM), Chinese Academy of Sciences, 310022 Hangzhou, China. [2]Key Laboratory of Prevention, Diagnosis and Therapy of Upper Gastrointestinal Cancer of Zhejiang Province, 310022 Hangzhou, China. [3]Zhejiang Provincial Research Center for Upper Gastrointestinal Tract Cancer, Zhejiang Cancer Hospital, 310022 Hangzhou, China. [4]State Key Laboratory of Systems Medicine for Cancer, School of Biomedical Engineering, Institute of Medical Robotics and Med-X Research Institute, Shanghai Jiao Tong University, 200030 Shanghai, China. [5]Division of Cardiology, Renji Hospital, School of Medicine, Shanghai Jiao Tong University, 200127 Shanghai, China. [6]Department of Epidemiology, School of Public Health, Zhejiang Chinese Medical University, 310053 Hangzhou, China. [7]Department of Gastric Surgery, Zhejiang Cancer Hospital, Hangzhou Institute of Medicine (HIM), Chinese Academy of Sciences, 310022 Hangzhou, China. [8]Office of Cancer Screening, National Cancer Center/National Clinical Research Center for Cancer/Cancer Hospital, Chinese Academy of Medical Sciences and Peking Union Medical College, 100021 Beijing, China. [9]School of Public Health and Management, Wenzhou Medical University, 325035 Wenzhou, China. [10]These authors contributed equally: Juan Zhu, Yida Huang, Xue Li, Bin Liu. ✉E-mail: Jayshu@bjmu.edu.cn; dulb@zjcc.org.cn; chengxd@zjcc.org.cn

and targeted therapy (Martínez-Reyes and Chandel, 2021; Pavlova et al, 2022).

Beyond their role as phenotypic markers, metabolites also play crucial biological roles. Growing evidence suggests that the metabolome interacts with and actively modulates all other omics levels, particularly the proteome (Rinschen et al, 2019; Piazza et al, 2018). For example, lactate, a product of glycolysis, can covalently modify histones through a process known as lactylation, thereby influencing histone activity and the expression of downstream genes (Zhang et al, 2019). Other metabolites responsible for post-translational modifications include succinyl-CoA (arginine succinylation), as well as activated sugar molecules (e.g., UDP-glucose) for glycosylation and GlcNAcylation (Dennis and Brewer, 2013; Hart et al, 2011). Beyond covalent modifications, metabolites can also directly interact with proteins in a non-covalent manner, impacting their functional activity. For instance, oleic acid can regulate the PI3K/Akt pathway by interacting with ENO1 on the cell membrane, thereby promoting colorectal cancer progression (Sun et al, 2024).

Isovalerylcarnitine (C5), a product derived from L-leucine catabolism, has been previously identified as under-expressed in GC patients and associated with a reduced risk of GC (Tsai et al, 2018). However, the specific roles and mechanisms of isovalerylcarnitine (C5) in GC remain poorly understood and warrant further investigation.

In this study, we conducted a multi-center, large-scale investigation and employed the mainstream untargeted ultra-performance liquid chromatography-mass spectrometry (UPLC-MS) metabolomics combined with machine learning to develop a non-invasive diagnostic model for GC. Further, we mechanistically explored the role of isovalerylcarnitine (C5) in GC progression, invasion, and metastasis, providing potential targets for further precision treatment and intervention of GC (Fig. 1).

# Results

## Characteristics of study participants

Schematic overview of the study design is presented in Fig. 1. A total of 597 participants were retrospectively enrolled. The discovery dataset comprised 144 individuals (mean age 63.18 ± 7.09 years; GC cases: 50, Non-GC: 94), and the validation dataset included 453 individuals from seven centers (mean age 62.51 ± 8.40 years; GC cases: 227, Non-GC: 226). 48.4% GC patients were diagnosed at stage I–II. No statistically significant differences regarding age, sex, smoking status, or *H. pylori* infection status were observed between GC and non-GC controls in either dataset (Table 1).

## Identification and validation of plasma metabolic biomarkers of gastric cancer

In the discovery dataset, partial least squares discriminant analysis (PLS-DA) demonstrated clear separation between GC and non-GC samples (Fig. 2A). We identified 93 candidate differential blood metabolites between GC and non-GC (Fig. 2B), of which 26 (10 upregulated, 16 downregulated) were validated in the independent dataset (Fig. 2C). Lipids and amino acids comprised 69.2% of these 26 metabolites (Fig. 2D), which were predominantly enriched in ketone body, methylhistidine, β-alanine, fatty acid, and histidine

pathways (Appendix Table S1; Appendix Fig. S1). The area under the curve (AUC) for individual metabolites ranged from 0.631 to 0.855 and from 0.548 to 0.895 in the discovery and validation dataset, with fold changes (FCs) and false discovery rates (FDRs) detailed in Appendix Table S2. Using a lasso-regression score (LRScore)-based approach (Jerome Friedman et al, 2010), we evaluated model AUCs across different LRScore thresholds and observed that AUCs plateaued until a threshold of 0.87, after which they declined sharply (Fig. 2E; Appendix Table S3).

## Development and validation of a metabolic panel for gastric cancer diagnosis

We applied five machine learning algorithms for model construction: neural network (NN), support vector machine (SVM), ridge regression (RR), lasso regression (LR), and naive bayes (NB). These 26 metabolites achieved AUCs of 0.898–0.971 (discovery) and 0.908–0.950 (validation), with sensitivity ranging from 0.800–0.940 and 0.811–0.903, specificity from 0.840–0.968 and 0.810–0.903, and accuracy from 0.826–0.931 and 0.843–0.898, respectively (Fig. 3A; Appendix Tables S4 and 5). Based on the LRScore-based approach, six metabolite panel—2,6-dihydroxybenzoic acid, cysteine s-sulfate, isovalerylcarnitine (C5), glycoursodeoxycholate, 2-hydroxy-3-methylvalerate, and N-acetylneuraminate—achieved the optimal AUC performance. We comprehensively assessed the performance of the six-metabolite panel, AUC ranged from 0.947–0.982 (discovery) and 0.920–0.951(validation); sensitivity, 0.900–0.940 (discovery) and 0.863–0.925 (validation); specificity, 0.883–0.936 and 0.858–0.889; accuracy, 0.889–0.938 and 0.861–0.901 (Fig. 3B; Table 2; Appendix Table S6). The NN model [Met-NN(6)] performed best, achieving AUC of 0.982 (95% CI: 0.965–0.998) in discovery and 0.951 (95% CI: 0.931–0.970) in validation dataset. We also confirmed that there was no overfitting of the model, based on the permutation test ($p < 0.001$, Appendix Fig. S2).

Met-NN(6) outperformed individual metabolite (Fig. 3C; Table 3; Appendix Fig. S3) and traditional risk factors (Fig. 3D), and decision-curve analysis showed superior net benefit compared to conventional predictors (Fig. 3E). Importantly, the sensitivity of Met-NN(6) (0.900–0.940 in discovery; 0.863–0.925 in validation) far exceeded that of standard blood biomarkers (0.020–0.240; 0.031–0.148) (Fig. 3F; Appendix Table S7). In particular, Met-NN(6) identified 70.0% and 65.2% of GC cases missed by CA72-4 alone in the discovery and validation sets (Fig. 3G). Sensitivity was highest in females (0.922) and in subjects younger than 55 years (0.942) (Fig. 3H). Finally, for early GC detection, the six-metabolite panel achieved sensitivity of 0.737–0.947 and 0.783–0.904, specificity of 0.819–1.000 and 0.872–0.889, accuracy of 0.841–0.973 and 0.842–0.889, and AUC of 0.914–0.961 and 0.894–0.940 in discovery and validation datasets (Fig. 3I; Table 2). After adjusting for age, sex, smoking, and drinking, Met-NN(6) retained high AUC of 0.980 (95% CI: 0.963–0.997) and 0.928 (95% CI: 0.904–0.952) in discovery and validation, respectively (Fig. 3J; Appendix Table S8).

## Validation of metabolic biomarkers in gastric cancer progression

To further investigate the potential roles of the aforementioned six metabolites in GC progression, we conducted metabolomic profiling in tumor tissues and matched adjacent non-tumor tissues

## Plasma metabolic biomarkers

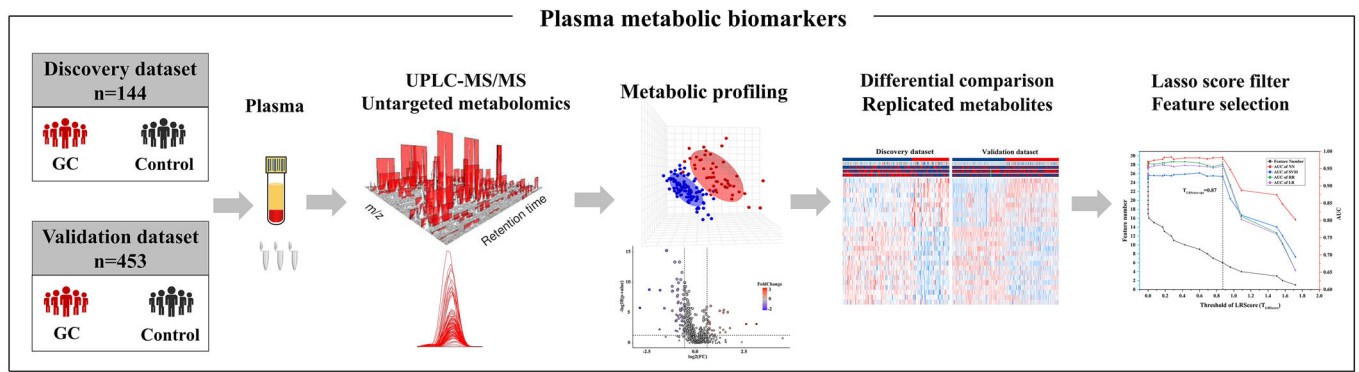

## Diagnostic model construction and validation

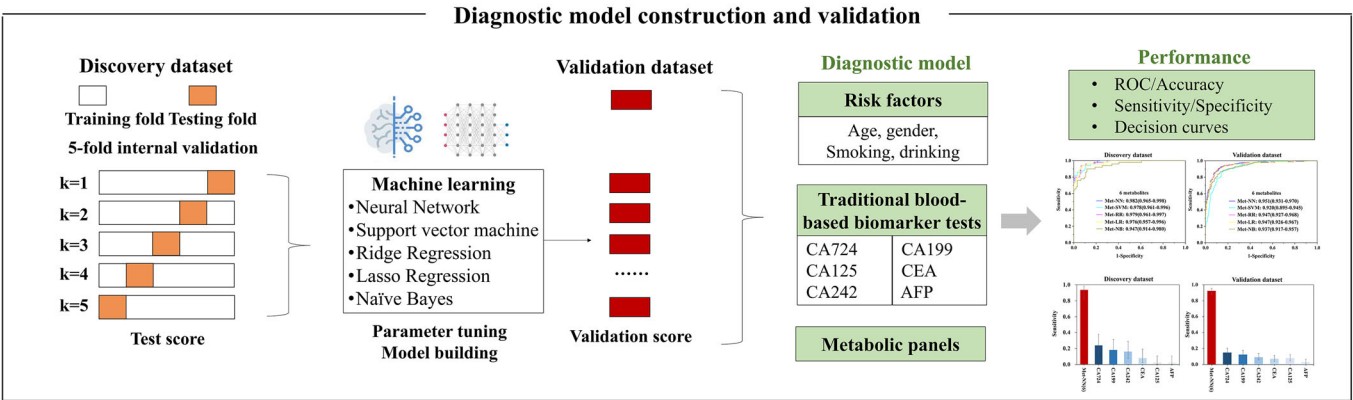

## Tissue validation

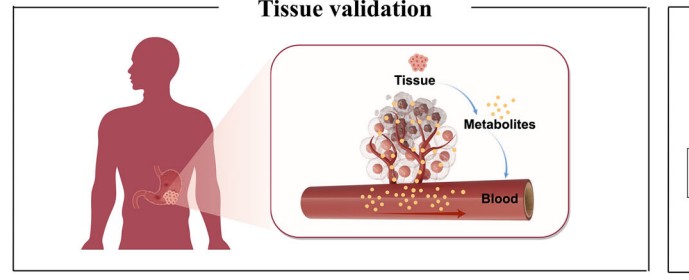

## MR validation

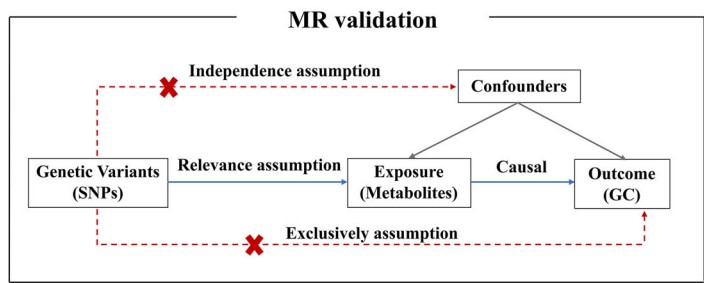

## Proteomic profiling

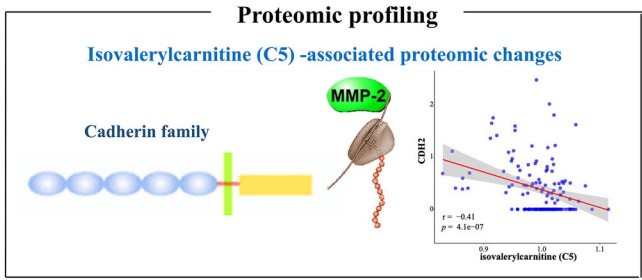

## Biological validation

## Clinical prognosis

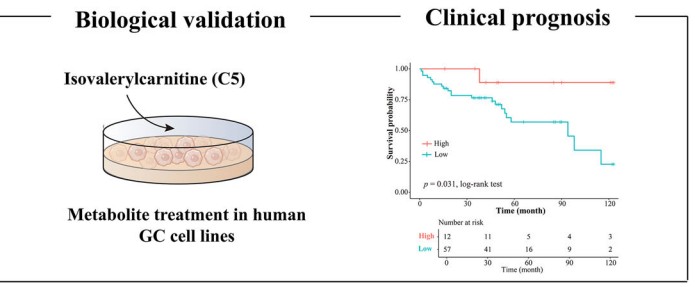

from 50 GC patients in the initial discovery dataset. The results revealed that two metabolites, isovalerylcarnitine (C5) and N-acetylneuraminate, were differentially expressed between GC tissues and paired adjacent tissues (Fig. 4A). Isovalerylcarnitine (C5) was significantly reduced, whereas N-acetylneuraminate was elevated in both plasma and tumor tissues of GC patients compared with controls (Fig. 4B).

To assess causality, bidirectional Mendelian randomization (MR) showed that genetically predicted isovalerylcarnitine (C5)

levels were inversely associated with GC risk (inverse-variance weighted, OR = 0.833, 95% CI: 0.697–0.995; P = 0.044), while N-acetylneuraminate showed no statistically significant association (P = 0.819) (Fig. 4C–E; Appendix Tables S9, 10, and 11). In sensitivity analyses, the maximum likelihood method produced similar estimates. Reverse MR did not reveal a statistically significant correlation between GC and isovalerylcarnitine (C5) (inverse-variance weighted: OR = 1.022; 95% CI: 0.985–1.060; P = 0.256) (Appendix Table S12).

◀ **Figure 1. Schematic overview of study design.**

The study included a total of 597 participants, whose plasma samples were analyzed using UPLC-MS/MS untargeted metabolomics. The discovery dataset comprised 144 participants, while the validation dataset included 453 participants. In the discovery phase, metabolic profiling and differential comparison identified 93 candidate differential blood metabolites between GC patients and non-GC controls. Among these, 26 replicated differential metabolites were validated in the external validation dataset. Feature selection using the Lasso score filter resulted in six key features, which were used to construct diagnostic models through machine learning algorithms, including neural network (NN), support vector machine (SVM), ridge regression (RR), logistic regression (LR), and naive Bayes (NB). The diagnostic performance of the model was evaluated for both GC and early-stage GC, with a focus on comparing its sensitivity to that of six commonly employed clinical blood tumor biomarkers (CA724, CA199, CA125, CEA, CA242, and AFP). Furthermore, the identified metabolic biomarkers were validated at tissue level. Isovalerylcarnitine (C5) and N-acetylneuraminate were validated as differentially expressed between GC tissues and paired normal adjacent tissues (NATs). To assess the potential causal relationship, Mendelian randomization (MR) analysis was performed. Isovalerylcarnitine (C5) was causally associated with a reduced risk of GC. Additionally, proteomic analysis on the same cohort's plasma samples revealed significant correlations between isovalerylcarnitine (C5) and cadherin and MMP families. Finally, in vitro experiments demonstrated that isovalerylcarnitine (C5) inhibited GC cell proliferation, migration and invasion by downregulating the expression of N-cadherin and MMP2.

**Table 1. Characteristics of GC and non-GC patients in the discovery and validation dataset.**

| | Discovery dataset | | | | Validation dataset | | | |
|---|---|---|---|---|---|---|---|---|
| | Overall (N = 144) | GC (n = 50) | Non-GC (n = 94) | P | Overall (N = 453) | GC (n = 227) | Non-GC (n = 226) | P |
| **Age, year** | 63.18 ± 7.09 | 63.66 ± 8.47 | 62.93 ± 6.27 | 0.591 | 62.51 ± 8.40 | 62.46 ± 10.52 | 62.55 ± 5.51 | 0.909 |
| **Age, year** | | | | 0.530 | | | | <0.001 |
| <55 | 22 (15.28) | 8 (16.00) | 14 (14.89) | | 71 (15.67) | 50 (22.03) | 21 (9.29) | |
| 55–64 | 49 (34.03) | 14 (28.00) | 35 (37.23) | | 189 (41.72) | 69 (30.40) | 120 (53.10) | |
| ≥65 | 73 (50.69) | 28 (56.00) | 45 (47.87) | | 193 (42.60) | 108 (47.58) | 85 (37.61) | |
| **Gender** | | | | 0.979 | | | | 0.637 |
| Male | 101(70.14) | 35 (70.00) | 66 (70.21) | | 309 (68.21) | 152 (66.96) | 157 (69.47) | |
| Female | 43 (29.86) | 15 (30.00) | 28 (29.79) | | 144 (31.79) | 75 (33.04) | 69 (30.53) | |
| **Smoking** | | | | 0.003 | | | | 0.059 |
| No | 79 (54.86) | 36 (72.00) | 43 (45.74) | | 292 (64.46) | 154 (67.84) | 138 (61.06) | |
| Yes | 65 (45.14) | 14 (28.00) | 51 (54.26) | | 157 (34.66) | 73 (32.16) | 84 (37.17) | |
| Unknown | | | | | 4 (0.88) | 0 (0.00) | 4 (1.77) | |
| **Drinking** | | | | <0.001 | | | | 0.093 |
| No | 62 (43.06) | 34 (68.00) | 28 (29.79) | | 334 (73.73) | 165 (72.69) | 169 (74.78) | |
| Yes | 81 (56.25) | 16 (32.00) | 6 5 (69.15) | | 115 (25.39) | 62 (27.31) | 53 (23.45) | |
| Unknown | 1 (0.69) | 0 (0.00) | 1 (1.06) | | 4 (0.88) | 0 (0.00) | 4 (1.77) | |
| **H. pylori** | | | | 0.938 | | | | – |
| Negative | 80 (55.56) | 28 (56.00) | 52 (56.52) | | – | – | – | |
| Positive | 64 (44.44) | 22 (44.00) | 42 (45.65) | | – | – | – | |
| **Stage**[a] | | | | – | | | | – |
| I | | 4 (8.00) | – | | | 63 (28.00) | – | |
| II | | 15 (30.00) | – | | | 52 (23.11) | – | |
| III | | 30 (60.00) | – | | | 100 (44.44) | – | |
| IV | | 1 (2.00) | – | | | 10 (4.44) | – | |
| **Clinical protein biomarker** | | | | | | | | |
| CA724 (U/ml) | – | 7.36 ± 20.25 | – | | – | 8.43 ± 33.27 | 10.39 ± 41.39 | 4.99 ± 5.18 | 0.064 |
| CA199 (U/ml) | – | 73.38 ± 250.52 | – | | – | 60.98 ± 665.28 | 89.55 ± 829.15 | 9.84 ± 7.98 | 0.153 |
| CA242 (U/ml) | – | 30.82 ± 85.00 | – | | – | 11.24 ± 57.55 | 15.21 ± 73.05 | 4.94 ± 5.50 | 0.042 |
| CA125 (U/ml) | – | 14.13 ± 7.67 | – | | – | 14.76 ± 13.36 | 16.06 ± 16.00 | 12.60 ± 6.50 | 0.007 |
| CEA (ng/ml) | – | 2.14 ± 2.56 | – | | – | 2.42 ± 7.80 | 2.93 ± 9.95 | 1.63 ± 0.96 | 0.055 |
| AFP (ng/ml) | – | 6.38 ± 27.07 | – | | – | 10.93 ± 89.56 | 16.11 ± 113.79 | 2.63 ± 1.10 | 0.078 |

*GC* gastric cancer, *CA724* carbohydrate antigen 724, *CA199* carbohydrate antigen 199, *CA242* carbohydrate antigen 242, *CA125* carbohydrate antigen 125, *CEA* carcinoembryonic antigen, *AFP* alpha-fetoprotein.
[a] There were two cases with unknown staging in the validation set.

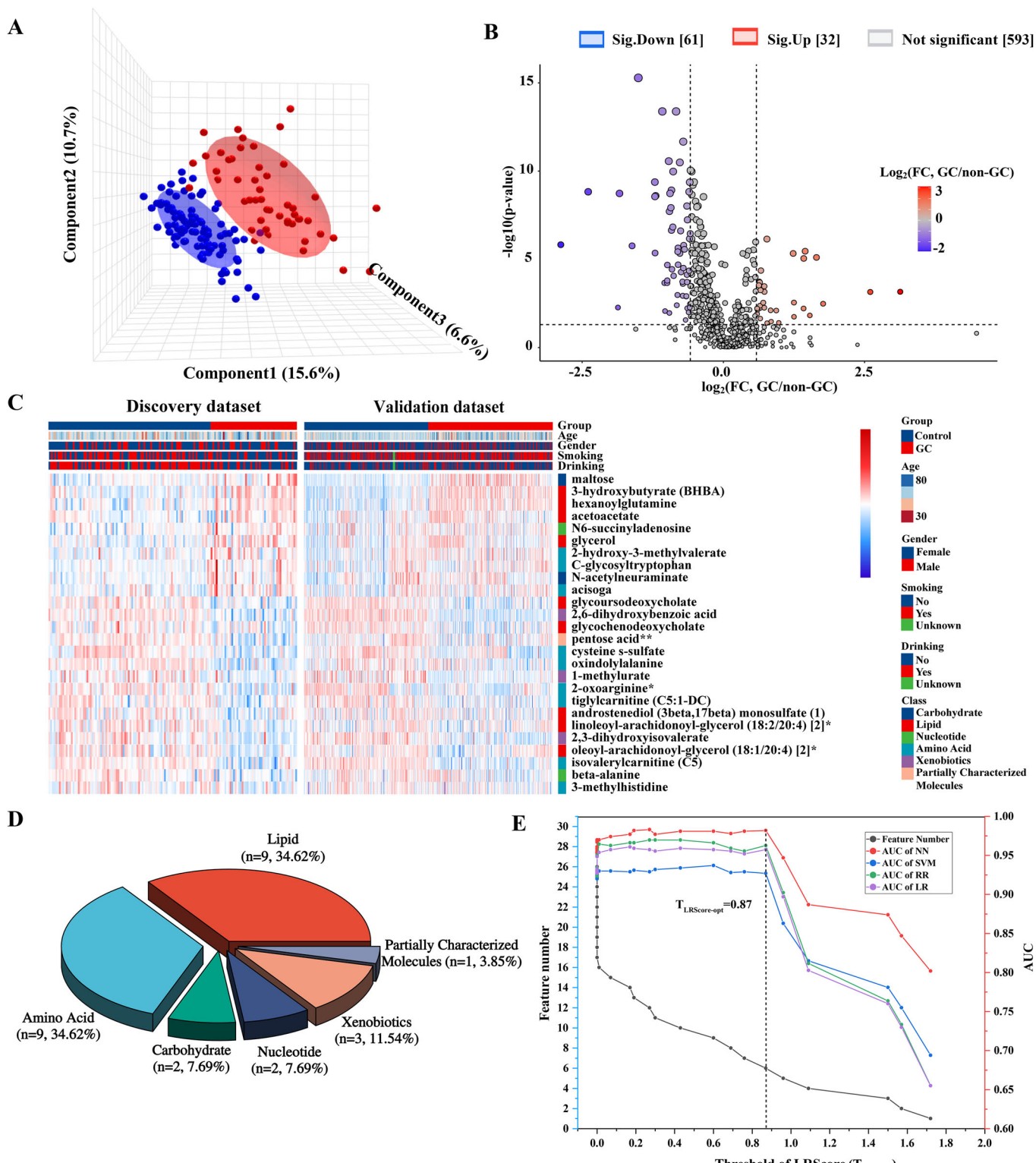

## Isovalerylcarnitine (C5)-associated proteomic changes revealed dysregulation of cell adhesion pathway in gastric cancer

To investigate the mechanisms by which isovalerylcarnitine (C5) inhibits GC progression and to identify its potential target proteins, proteomic analysis of plasma from 50 GC patients and 94 controls in the discovery dataset identified 656 proteins whose levels significantly correlated with isovalerylcarnitine (C5) ($P < 0.05$, Appendix Table S13). Of these, 44 proteins exhibited moderately strong correlations ($|r| > 0.4$, Table 4). Pathway enrichment analysis of these 44 proteins revealed significant enrichment in cell

**Figure 2.  Identification and validation of plasma metabolic biomarkers of gastric cancer.**

(A) Partial least squares discriminant analysis (PLS-DA) three-dimensional scores plot distinguishing GC patients from non-GC controls. $N = 16$ technical replicates. (B) Volcano plot illustrating differential metabolites between GC patients and non-GC controls in the discovery dataset. In the discovery phase, metabolic profiling and differential comparison identified 93 candidate differential blood metabolites between GC patients and non-GC controls. Statistical analyses were performed using Welch's $t$ test. (C) 26 replicated differential metabolites were validated in the external validation dataset. Heatmap displaying the expression of 26 metabolites in both discovery and validation datasets. (D) Pie chart representing the categories of 26 differentially repeated metabolites. (E) Area under the curve (AUC) values generated by machine learning algorithms, based on varying numbers of signals (grey line) and different thresholds of lasso regression score (LRScore, upper lines). The LR algorithm achieved the highest AUC at an optimized LRScore threshold (TLRScore) of 0.87 when using six metabolite features. Source data are available online for this figure.

adhesion and extracellular matrix organization processes (Fig. 5A). Notably, several cadherin (CDH) family members (CDH2, CDH5, CDH11, CDH13, CDHR5, PCDH1, and PCDH12) and matrix metalloproteinase (MMP) family members (MMP2 and MMP19) were negatively correlated with isovalerylcarnitine (C5) levels (Fig. 5B,C). Given that the dysregulation of the cell adhesion pathway is associated with tumor invasion and metastasis, it is hypothesized that isovalerylcarnitine (C5) may inhibit the invasion and metastasis of GC cells.

## Isovalerylcarnitine (C5) suppresses gastric cancer metastasis via regulation of VE-cadherin and MMP2

To further validate the effects of isovalerylcarnitine (C5) on the invasion and metastasis of GC cells, in vitro treatment of GC cells with isovalerylcarnitine (C5) significantly inhibited cell migration and invasion, as demonstrated by wound-healing and transwell assays (Fig. 6A–C). Western blotting revealed that isovalerylcarnitine (C5) markedly reduced CDH5 (VE-cadherin) and MMP2 expression (Fig. 6D,E), key mediators of GC metastasis, consistent with proteomic data showing their upregulation in GC (Fig. 6F).

Considering that isovalerylcarnitine (C5) has been identified as an activator of calpain, a calcium-dependent protease, we hypothesized that isovalerylcarnitine (C5) regulates VE-cadherin and MMP2 through calpain-mediated degradation. Consistent with this hypothesis, we observed that isovalerylcarnitine (C5) dose-dependently enhanced calpain activity in GC cells (Fig. 7A). To further elucidate calpain's role in VE-cadherin and MMP2 regulation, we mimicked calpain activation in vitro by supplementing calcium ions and assessed its impact on protein stability. Calcium treatment significantly reduced VE-cadherin and MMP2 levels, an effect that was abolished by calpeptin, a specific calpain inhibitor (Fig. 7B), supporting the notion that calpain activation facilitates their proteolytic degradation.

Critically, at the cellular level, calpeptin treatment reversed the isovalerylcarnitine (C5)-induced downregulation of VE-cadherin and MMP2 (Fig. 7C,D). Furthermore, the anti-metastatic effects of isovalerylcarnitine (C5) were also negated by calpeptin (Fig. 7E–G). Collectively, these findings demonstrate that isovalerylcarnitine (C5) inhibits GC metastasis by activating calpain, which in turn promotes the degradation of VE-cadherin and MMP2.

## Isovalerylcarnitine (C5) is negatively correlated with metastasis of GC patients

We further analyzed the relationship between isovalerylcarnitine (C5) and cancer metastasis in 277 GC patients from the discovery and validation datasets. Plasma isovalerylcarnitine (C5) levels

showed a significant inverse relationship with disease progression: concentrations declined progressively with higher stage and greater lymph node involvement (Fig. 8A,B) and were markedly lower in patients exhibiting distant metastasis (Fig. 8C). Given the association between distant spread and poor prognosis, we next examined the potential prognostic value of isovalerylcarnitine (C5) in a subset of 69 patients with follow-up. Consistent with our hypothesis, lower baseline C5 levels were significantly linked to inferior clinical outcomes (Fig. 8D), supporting its potential utility as a prognostic biomarker in GC management.

# Discussion

In this multi-phase study, we identified circulating metabolic biomarkers in the plasma distinguishing GC and non-GC patients using a non-invasive approach and developed a highly sensitive GC diagnostic model using machine learning algorithms. Our metabolic panel far outperformed conventional blood-based protein biomarkers in GC diagnosis. Moreover, we pinpointed isovalerylcarnitine (C5) as a potential metabolite closely associated with the progression of GC and comprehensively demonstrated its significant roles in inhibiting the migration and invasion of GC cells. Collectively, our research not only offers novel perspectives on the metabolic regulatory mechanisms underlying GC progression but also presents a promising biomarker-based strategy that holds great potential for clinical translation.

Blood, the most commonly used liquid biopsy sample, is characterized by its simple collection procedures, easy interpretation of test results, high acceptability for routine testing, and rich content of biomolecules (Ma et al, 2023). In terms of genetic molecular markers, various circulating nucleic acids in the bloodstream have been recognized as indicative biomarkers of GC, such as microRNA, cell-free RNA, and cell-free DNA (So et al, 2021; Li et al, 2024b; Wang et al, 2024; Tao et al, 2023). Regarding proteomic biomarkers, the clinical applications of cancer-associated antigens such as CEA, AFP, CA199, and CA125 have been extensively researched (Leja and Linē, 2021), yet their effectiveness remains constrained owing to low sensitivity and accuracy, typically around 30%. In contrast, metabolic biomarkers, representing terminal products of biochemical cascades, are poised to deliver real-time physiological snapshots of biological systems and hold promise as highly efficacious therapeutic targets for intervention purposes (Su et al, 2021; Drew et al, 2021). Before disease onset or during accumulation of risk factors, endogenous molecules elicit corresponding metabolic responses, enabling a more sensitive identification of specific metabolic phenotypes resulting from the interaction of genetic and environmental factors (Buergel et al, 2022; Schmidt et al, 2021). Moreover, the number of metabolites is much

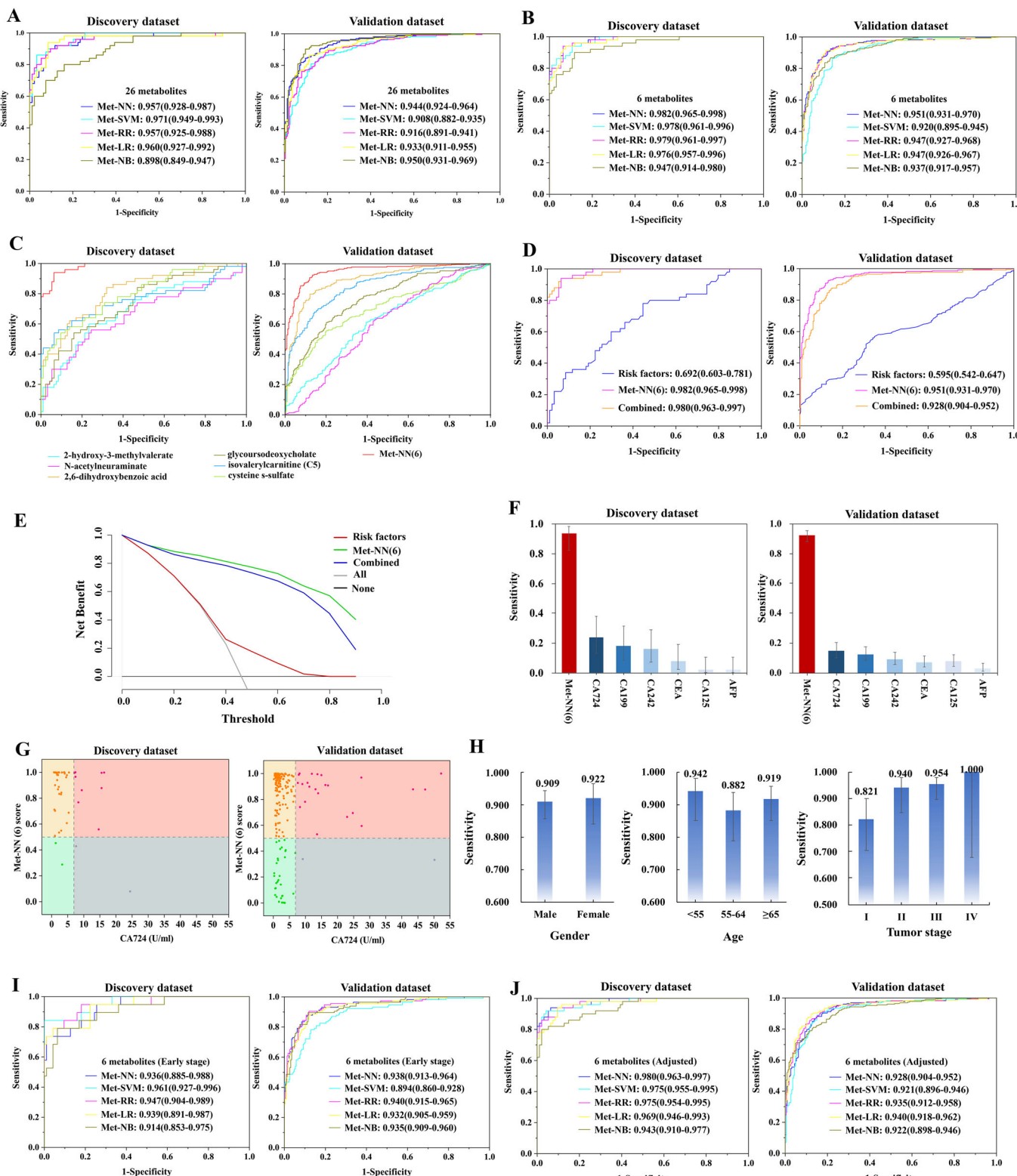

smaller than that of corresponding genes and proteins, which enhances detection of minuscule alterations in gene and protein expression at the metabolic level, with enhanced sensitivity in detecting minor alterations in metabolic characteristics. The widespread application of high-throughput sequencing and high-sensitivity mass

spectrometry technologies has significantly enhanced the detection capabilities in blood metabolomics and the potential for discovering novel biomarkers. UPLC-MS/MS is a mainstream technique in metabolomics analysis, with advantages such as high sensitivity, high resolution, high throughput, and deep coverage. Given the strong

**Figure 3. Development and validation of a metabolic panel for gastric cancer diagnosis.**

(A) We applied five machine learning algorithms for model construction: neural network (NN), support vector machine (SVM), ridge regression (RR), lasso regression (LR), and naive bayes (NB). Receiver operating characteristic (ROC) curves of 26 metabolites through machine learning algorithms. The area under the curve (AUC) ranges from 0.898 to 0.971 in the discovery dataset and from 0.908 to 0.950 in the validation dataset. (B) The AUC of the six-metabolite panel ranges from 0.947 to 0.982 in the discovery dataset and from 0.920 to 0.951 in the validation dataset. (C) ROC curves of the metabolic panel demonstrate improved AUC values of 0.982 (95% CI: 0.965–0.998, $P < 0.05$, DeLong test) in the discovery dataset and 0.951 (95% CI: 0.931–0.970, $P < 0.05$, DeLong test) in the validation dataset, compared to individual metabolic biomarkers with AUCs of 0.661–0.815 and 0.595–0.895, respectively. (D) ROC curves of risk factors, the metabolic panel, and the combined panel. (E) Decision curves analysis evaluating the net benefit of risk factors, the metabolic panel, and the combined panel. (F) Sensitivity of the metabolic panel and clinical blood-based biomarker tests for carbohydrate antigen 724 (CA724), carbohydrate antigen 199 (CA199), carbohydrate antigen 242 (CA242), carcinoembryonic antigen (CEA), carbohydrate antigen 125 (CA125), and alpha-fetoprotein (AFP) in both the discovery (sample number, $N = 50$) and validation datasets (sample number, $N = 453$). Data were presented with sensitivity (95% CI). (G) Scatter plot illustrating metabolic panel scores and CA724 levels in GC patients from both discovery and validation datasets. (H) Sensitivity of GC patients stratified by sex (sample number, $N = 597$), age (sample number, $N = 597$), and tumor stage (sample number, $N = 275$). Data were presented with sensitivity (95% CI). (I) ROC curves of the metabolic panel specifically for early-stage (stages 0–II) GC patients, with AUC of 0.914–0.961 and 0.894–0.940 in the discovery and validation dataset, respectively. (J) After adjusting for age, sex, smoking, and drinking, Met-NN(6) retained high AUC of 0.980 (95% CI: 0.963–0.997) and 0.928 (95% CI: 0.904–0.952) in discovery and validation, respectively. Source data are available online for this figure.

**Table 2. The diagnostic performance of six metabolic biomarkers panel in the discovery and validation dataset.**

| Machine learning | Performance | GC (all stage) | | GC (early stage) | |
|---|---|---|---|---|---|
| | | Discovery dataset | Validation dataset | Discovery dataset | Validation dataset |
| NN | Sensitivity (95% CI) | 0.940 (0.825–0.984) | 0.925 (0.881–0.954) | 0.737 (0.486–0.899) | 0.878 (0.801–0.929) |
| | Specificity (95% CI) | 0.936 (0.861–0.974) | 0.867 (0.814–0.907) | 0.989 (0.934–0.999) | 0.889 (0.839–0.926) |
| | Accuracy (95% CI) | 0.938 (0.881–0.969) | 0.896 (0.864–0.922) | 0.947 (0.883–0.978) | 0.886 (0.846–0.916) |
| | AUC (95% CI) | 0.982 (0.965–0.998) | 0.951 (0.931–0.970) | 0.936 (0.885–0.988) | 0.938 (0.913–0.964) |
| SVM | Sensitivity (95% CI) | 0.940 (0.825–0.984) | 0.863 (0.810–0.904) | 0.842 (0.595–0.958) | 0.783 (0.694–0.852) |
| | Specificity (95% CI) | 0.894 (0.809–0.945) | 0.858 (0.804–0.900) | 1.000 (0.951–1.000) | 0.872 (0.819–0.911) |
| | Accuracy (95% CI) | 0.910 (0.848–0.949) | 0.861 (0.825–0.891) | 0.973 (0.919–0.993) | 0.842 (0.798–0.878) |
| | AUC (95% CI) | 0.978 (0.961–0.996) | 0.920 (0.895–0.945) | 0.961 (0.927–0.996) | 0.894 (0.860–0.928) |
| RR | Sensitivity (95% CI) | 0.940 (0.825–0.984) | 0.907 (0.860–0.940) | 0.947 (0.719–0.997) | 0.904 (0.832–0.949) |
| | Specificity (95% CI) | 0.936 (0.861–0.974) | 0.885 (0.834–0.922) | 0.819 (0.723–0.888) | 0.881 (0.829–0.918) |
| | Accuracy (95% CI) | 0.938 (0.881–0.969) | 0.896 (0.864–0.922) | 0.841 (0.757–0.900) | 0.889 (0.849–0.919) |
| | AUC (95% CI) | 0.979 (0.961–0.997) | 0.947 (0.927–0.968) | 0.947 (0.904–0.989) | 0.940 (0.915–0.965) |
| LR | Sensitivity (95% CI) | 0.940 (0.825–0.984) | 0.912 (0.865–0.944) | 0.789 (0.539–0.930) | 0.878 (0.801–0.929) |
| | Specificity (95% CI) | 0.926 (0.848–0.967) | 0.889 (0.839–0.926) | 0.957 (0.888–0.986) | 0.881 (0.829–0.918) |
| | Accuracy (95% CI) | 0.931 (0.873–0.964) | 0.901 (0.868–0.926) | 0.929 (0.861–0.967) | 0.880 (0.839–0.911) |
| | AUC (95% CI) | 0.976 (0.957–0.996) | 0.947 (0.926–0.967) | 0.939 (0.891–0.987) | 0.932 (0.905–0.959) |
| NB | Sensitivity (95% CI) | 0.900 (0.774–0.963) | 0.868 (0.815–0.908) | 0.789 (0.539–0.930) | 0.896 (0.821–0.943) |
| | Specificity (95% CI) | 0.883 (0.796–0.937) | 0.867 (0.814–0.907) | 0.936 (0.861–0.974) | 0.885 (0.834–0.922) |
| | Accuracy (95% CI) | 0.889 (0.823–0.933) | 0.868 (0.832–0.897) | 0.912 (0.839–0.954) | 0.889 (0.849–0.919) |
| | AUC (95% CI) | 0.947 (0.914–0.980) | 0.937 (0.917–0.957) | 0.914 (0.853–0.975) | 0.935 (0.909–0.960) |

*GC* gastric cancer, *NN* neural network, *SVM* support vector machine, *RR* ridge regression, *LR* lasso regression, *NB* naive bayes, *AUC* area under the curve, *CI* confidence interval.

correlation between GC and metabolic processes, our study underscores the predictive potential of GC-related metabolites.

Compared with recent state-of-the-art metabolomics-based models (Chen et al, 2024a; Huang et al, 2021; Xu et al, 2023), our study utilized two independent MS analytical platforms, demonstrated consistently robust performance across both discovery and validation cohorts, highlighting cross-cohort generalizability. Additionally, we incorporated Mendelian randomization and biological validation of the key metabolites, further strengthening the biological plausibility of our findings. These results not

only provide a clinically relevant diagnostic model but also offer novel mechanistic insights into gastric cancer progression.

Traditional analytical methods are ineffective at handling high-dimensional data. In contrast, machine learning algorithms are powerful tools for elucidating the intricate associations between omics data and disease status as well as constructing predictive models (Xu et al, 2023; Chen et al, 2024a; Greener et al, 2022). We employed five machine learning algorithms. These algorithms efficiently managed large-scale datasets, thereby streamlining the data analysis process, aiding in dimensionality reduction, and

**Table 3.    Six metabolic biomarkers in the discovery and validation dataset.**

| No. | Metabolic biomarkers | Discovery dataset | | | Validation dataset | | |
|---|---|---|---|---|---|---|---|
| | | FC | FDR | AUC (95% CI) | FC | FDR | AUC (95% CI) |
| 1 | 2,6-dihydroxybenzoic acid | 0.43 | 4.04E-10 | 0.815 (0.745–0.885) | 0.29 | 3.57E-65 | 0.895 (0.865–0.924) |
| 2 | cysteine s-sulfate | 0.52 | 1.81E-09 | 0.796 (0.724–0.868) | 0.71 | 7.47E-17 | 0.702 (0.653–0.750) |
| 3 | isovalerylcarnitine (C5) | 0.64 | 9.68E-09 | 0.757 (0.680–0.833) | 0.46 | 1.90E-38 | 0.847 (0.812–0.883) |
| 4 | glycoursodeoxycholate | 0.33 | 1.73E-06 | 0.741 (0.658–0.824) | 0.37 | 1.66E-22 | 0.768 (0.725–0.811) |
| 5 | 2-hydroxy-3-methylvalerate | 1.66 | 6.76E-04 | 0.690 (0.601–0.778) | 1.20 | 1.20E-06 | 0.616 (0.564–0.667) |
| 6 | N-acetylneuraminate | 1.63 | 2.78E-03 | 0.661 (0.572–0.751) | 1.11 | 2.86E-03 | 0.595 (0.543–0.648) |

*FC* fold change, *FDR* false discovery rate, *AUC* area under the curve, *CI* confidence interval.

facilitating the selection of the most representative metabolites for distinguishing between GC and non-GC samples, thereby enhancing the robustness of the diagnostic model and reducing overfitting during training. Moreover, machine learning algorithms automatically identify the underlying associations among metabolites, revealing the latent rules within complex correlations. This provides novel insight into the complexity of biological systems and ultimately improves the diagnostic accuracy, generalization ability, and interpretability of disease diagnosis.

To advance the clinical translation of our findings, we plan to conduct absolute quantification metabolomics using a large-scale, multi-center patient cohort to establish reliable diagnostic thresholds for the key metabolites. In parallel, we are developing cost-effective detection platforms (e.g., ELISA, LC-MS/MS), with estimated per-test costs of USD 20–50, and will initiate a prospective validation study to assess diagnostic performance and facilitate regulatory approval. Based on the predictive model, commercial diagnostic kits and streamlined testing devices could be developed to enable early GC detection and guide clinical decision-making through individualized risk stratification. Furthermore, we aim to investigate the utility of plasma metabolite biomarkers in predicting GC risk and identifying high-risk individuals within asymptomatic populations, thereby supporting stratified screening and targeted intervention strategies.

In addition to serving as crucial indicators of phenotypes, metabolites can exert biological functions through various pathways, particularly via interactions with proteins. In our study, we employed tissue metabolomics validation and MR analysis to identify isovalerylcarnitine (C5) as a functionally significant metabolite potentially inhibiting GC progression. Through integrative correlation analysis of isovalerylcarnitine (C5) levels and proteomic data, we identified the cadherin and MMP families as key downstream molecular targets potentially regulated by isovalerylcarnitine (C5). We then demonstrated that isovalerylcarnitine (C5) induces downregulation of VE-cadherin and MMP2 through calpain activation.

VE-cadherin, a transmembrane glycoprotein predominantly expressed by vascular endothelial cells, belongs to the cadherin superfamily. Structurally, VE-cadherin consists of an extracellular domain mediating calcium-dependent homophilic cell adhesion, a transmembrane domain, and an intracellular domain that interacts with catenins to link with the cytoskeleton, thereby maintaining intercellular junction stability (Dejana et al, 2008; Lampugnani et al, 1995). Accumulating evidence suggests that highly aggressive

tumor cells can express VE-cadherin and form endothelial-like structures, participating in vasculogenic mimicry (VM). VM refers to the formation of vascular-like channels by tumor cells independent of endothelial cells, providing nutrients and oxygen to tumors. VE-cadherin facilitates intercellular connections among tumor cells, enabling the construction of functional vascular-like networks (Delgado-Bellido et al, 2017). Studies have demonstrated that tumor cell-expressed VE-cadherin contributes to VM network formation in cancers such as GC and non-small cell lung cancer, allowing tumors to sustain nutrient supply even in the absence of angiogenesis, thereby promoting tumor growth and metastasis (Li et al, 2017; Williamson et al, 2016). Our study reveals that isovalerylcarnitine (C5) regulates the expression of VE-cadherin and MMP2 in GC cells, representing a key mechanism underlying its inhibition of GC cell invasion and metastasis. On one hand, isovalerylcarnitine (C5) downregulates VE-cadherin, impairing the migratory capacity of GC cells. On the other hand, it suppresses MMP2 expression, reducing extracellular matrix degradation and further inhibiting invasive potential.

Our study uncovers a paradoxical duality in the role of L-leucine metabolism in GC progression. While L-leucine and other branched-chain amino acids (BCAAs) are well-established promoters of tumorigenesis (Choi et al, 2024), primarily via mTORC1 pathway activation to drive cancer cell proliferation, we demonstrate for the first time that its downstream metabolite, isovalerylcarnitine (C5) inhibits GC invasion and metastasis. This apparent contradiction may stem from metabolic bifurcation within the L-leucine catabolic pathway, where substrate flux partitioning determines oncogenic versus tumor-suppressive outcomes. Canonically, L-leucine is transaminated by branched-chain amino acid transferases (BCATs, notably BCAT1) to generate α-ketoisocaproate (2-KIC), which is subsequently decarboxylated by the branched-chain α-ketoacid dehydrogenase complex (BCKDC) to form isovaleryl-CoA. Under normal physiological conditions, isovaleryl-CoA is oxidized by isovaleryl-CoA dehydrogenase (IVD) to 3-methylcrotonyl-CoA, entering β-oxidation and ketogenesis to yield acetyl-CoA, thereby fueling energy production and biosynthetic processes that sustain tumor growth (Dimou et al, 2022). However, when IVD activity is compromised (e.g., due to genetic defects or mitochondrial dysfunction), isovaleryl-CoA accumulates and is diverted toward conjugation with carnitine, forming isovalerylcarnitine (C5) (Dimou et al, 2022), which we identify as a novel tumor suppressor in GC.

Notably, the elevated BCAT1 activity (Xu et al, 2018; Shu et al, 2021; Qian et al, 2023), while reduced isovalerylcarnitine (C5)

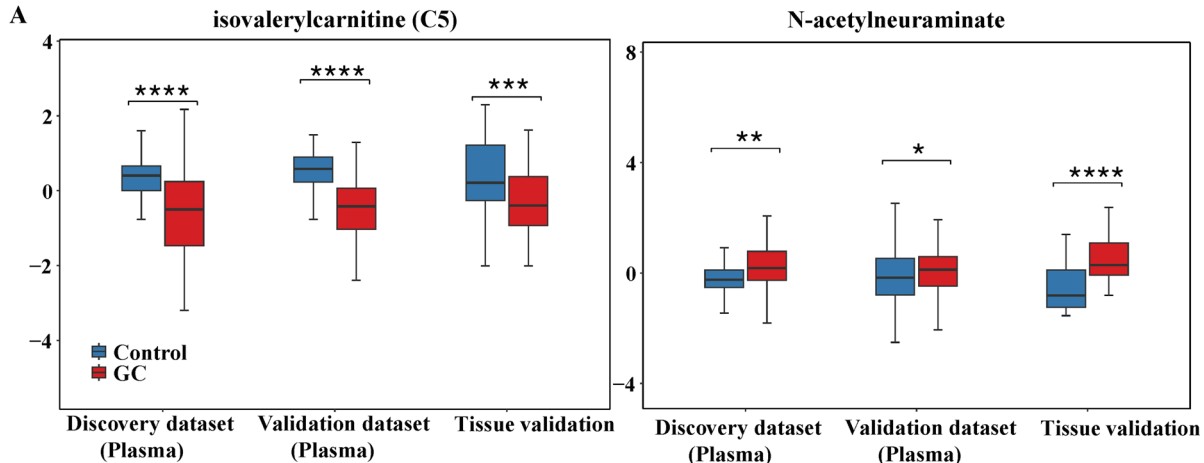

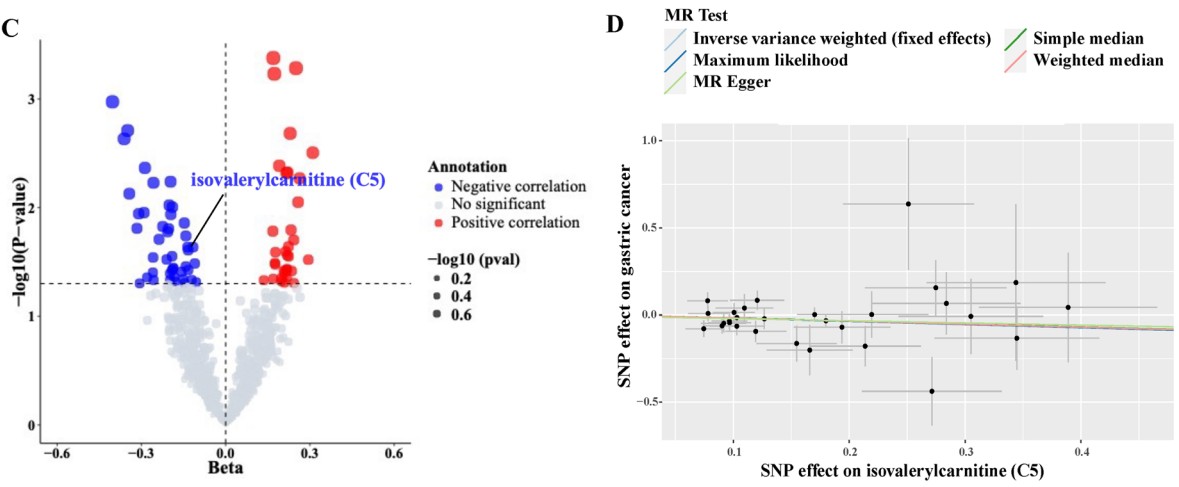

| Exposure_outcome | Methods | SNP | Beta | SE | OR | 95%CI | P | Hets | Pleios |
|---|---|---|---|---|---|---|---|---|---|
| **isovalerylcarnitine (C5)** | Inverse variance weighted (fixed effects) | 29 | -0.183 | 0.091 | 0.833 | 0.697-0.995 | 0.044 | 0.338 | |
| | Maximum likelihood | 29 | -0.181 | 0.092 | 0.834 | 0.696-0.999 | 0.049 | | |
| | Simple median | 29 | -0.170 | 0.141 | 0.843 | 0.64-1.111 | 0.226 | | |
| | Weighted median | 29 | -0.175 | 0.136 | 0.840 | 0.644-1.095 | 0.197 | | |
| | MR Egger | 29 | -0.123 | 0.265 | 0.885 | 0.526-1.487 | 0.647 | | 0.809 |
| **N-acetylneuraminate** | Inverse variance weighted (fixed effects) | 35 | 0.151 | 0.097 | 1.163 | 0.962-1.406 | 0.119 | 0.489 | |
| | Maximum likelihood | 35 | 0.154 | 0.099 | 1.167 | 0.96-1.417 | 0.121 | | |
| | Simple median | 35 | 0.028 | 0.151 | 1.029 | 0.765-1.382 | 0.852 | | |
| | Weighted median | 35 | 0.027 | 0.137 | 1.028 | 0.786-1.344 | 0.841 | | |
| | MR Egger | 35 | 0.021 | 0.206 | 1.021 | 0.682-1.529 | 0.920 | | 0.479 |

**Figure 4.  Validation of metabolic biomarkers in tissues and bidirectional Mendelian randomization analysis.**

(A) Among the six plasma metabolites in the panel, isovalerylcarnitine (C5) and N-acetylneuraminate were validated as differentially expressed between GC (sample number, $N = 50$) and paired adjacent tissues (sample number, $N = 50$). Two-tailed $t$ test, The exact adjusted $P$ values for isovalerylcarnitine (C5): discovery dataset (plasma) GC vs control $P < 0.0001$; validation dataset (plasma) GC vs control $P < 0.0001$; tissue validation GC vs control $P = 0.0006$. The exact adjusted $P$ values for N-acetylneuraminate: discovery dataset (plasma) GC vs control $P = 0.0071$; validation dataset (plasma) GC vs control $P = 0.019$; tissue validation GC vs control $P < 0.0001$.
$^*P < 0.05$, $^{**}P < 0.01$, $^{***}P < 0.001$, $^{****}P < 0.0001$. Box plots show the median (centre line), interquartile range (bounds of box, 25th–75th percentile), and whiskers extending to the minimum and maximum values. (B) The levels of isovalerylcarnitine (C5) were significantly reduced in both plasma and tumor tissues of GC patients, whereas N-acetylneuraminate was upregulated in both plasma and tumor tissues of GC patients. (C) Isovalerylcarnitine (C5) was further validated through Mendelian randomization (MR) analysis. Statistical analyses were performed using inverse-variance weighted analysis. (D) Scatterplot illustrating isovalerylcarnitine (C5) in the main analysis (inverse-variance weighted, IVW method) and several sensitivity analyses (sample number, $N = 288{,}444$). Data were 95% CI. (E) Circulating levels of isovalerylcarnitine (C5) were negatively associated with the risk of GC (IVW derived, $P < 0.05$). Source data are available online for this figure.

levels in GC, suggested that tumors may preferentially shunt L-leucine-derived carbon flux toward IVD-mediated oxidation rather than isovalerylcarnitine (C5) generation. This metabolic rewiring could reflect an adaptive mechanism to avoid isovalerylcarnitine (C5)-induced tumor suppression while sustaining mTORC1-driven proliferation. The downregulation of isovalerylcarnitine (C5) in GC patients further implies that IVD pathway hyperactivity may be a hallmark of aggressive disease, positioning IVD as a potential therapeutic target.

GC exhibits substantial global heterogeneity, largely attributable to geographic differences in *H. pylori* strain variations and dietary patterns. In East Asia—particularly China, Japan, and Korea—highly virulent *H. pylori* strains carrying cagA and vacA s1/m1 alleles are prevalent and strongly associated with elevated non-cardia GC risk (Azuma, 2004; Hooi et al, 2017; Tourrette et al, 2024). In contrast, Western strains often lack these high-risk genotypes, contributing to divergent GC burdens (Park et al, 2018). Moreover, dietary patterns also shape regional risk profiles(Bertuccio et al, 2013). In East Asia, particularly in Korea and China, the frequent consumption of salt-preserved and pickled foods introduces high levels of N-nitroso compounds and salt, which synergize with *H. pylori*-induced gastric injury to promote carcinogenesis (Fang et al, 2015; Wu et al, 2021). Conversely, in Western countries, increased intake of processed meats and low fiber consumption are recognized contributors to GC risk. These regional differences in *H. pylori* and dietary exposures underscore the importance of developing geographically tailored diagnostic and prevention strategies. While our model demonstrated robust performance in a multi-center Chinese cohort, further external validation in ethnically and geographically diverse populations is essential to ensure generalizability and clinical utility across global settings.

The novelty of this study is its comprehensive approach, which includes a large-scale analysis of highly sensitive metabolomics data from GC and non-GC individuals across multiple centers, thereby incorporating internal cross-validation and external validation. Integration of machine learning and metabolomics offers significant advantages, complementing various studies in GC characterization and precision medicine. Our Met-NN model, constructed from just six simple metabolites, allows for easy replication, optimization, and clinical implementation. Looking ahead, these models, with their assessment of the relative range of metabolites, can be employed for application in diverse scenarios, independent of specific tools and detection techniques. We compared the performance of the diagnostic model with six clinical protein biomarkers and identified potential causative differential metabolites through MR, providing insights into the disease etiology. Furthermore, we performed tissue-level tracing of plasma differential metabolites and explored molecular events that potentially drive GC development, providing insights into the metabolic reprogramming landscape of GC. This approach to tissue-based biomarker analysis would be instrumental in advancing the development of targeted therapeutic strategies, including metabolic and targeted therapies.

This study has some limitations. First, we exclusively focused on the Chinese population, warranting further investigations to verify its relevance and generalizability across populations from a wide array of geographic locales and ethnic backgrounds. Second, our study centered on GC diagnosis; future research should prioritize validating the diagnostic model in high-risk populations, such as those with intestinal metaplasia and intraepithelial neoplasia, to evaluate its performance in the early stages of disease progression. Third, the tissue metabolomics validation ($n = 50$ pairs) may be insufficiently powered for robust subgroup analyses, warranting larger-scale validation to explore intra-tumoral heterogeneity and clinical correlations. Fourth, while we identified isovalerylcarnitine (C5) as a tumor-suppressive metabolite, its functional role in GC pathogenesis requires further in vivo validation. Further studies using animal models are needed to confirm the impact of isovalerylcarnitine (C5) on gastric tumorigenesis and invasion, which would strengthen the rationale for its clinical translation. Finally, our MR analysis relied primarily on European-derived genetic instruments due to the scarcity of large-scale metabolomics GWAS in East Asian populations. Although supplementary analyses in Asian cohorts (e.g., Biobank Japan) supported the robustness of our findings, future large-scale GWAS in Chinese or East Asian populations are warranted to fully validate causal inferences and ensure the generalizability of our results across ancestries.

In conclusion, this study identified key blood metabolic biomarkers for GC diagnosis and developed a validated diagnostic model. These results highlight the cross-cohort generalizability of metabolite-based diagnostic approaches, underscoring the promise of metabolic profiling in precise GC screening and early diagnosis. Our study contributes significantly to the development of metabolite-based diagnostic models and advanced therapeutic strategies for GC.

**Table 4.** Differentially expressed proteins significantly correlated with isovalerylcarnitine (C5).

| Proteins | |r| > 0.4 | P value |
|---|---|---|
| QSOX1 | −0.447 | 1.89E-08 |
| AQR | −0.438 | 4.11E-08 |
| PIP5K1C | −0.413 | 2.70E-07 |
| PAPLN | −0.444 | 2.44E-08 |
| EFEMP2 | −0.460 | 6.74E-09 |
| FN1 | −0.496 | 2.64E-10 |
| SHBG | −0.501 | 1.66E-10 |
| PROS1 | −0.436 | 4.60E-08 |
| ASGR2 | −0.438 | 4.14E-08 |
| CTSB | −0.455 | 1.01E-08 |
| MMP2 | −0.419 | 1.68E-07 |
| CD14 | −0.506 | 9.82E-11 |
| VCAN | −0.567 | 1.31E-13 |
| ACP3 | −0.411 | 3.10E-07 |
| ELN | −0.476 | 1.61E-09 |
| CDH2 | −0.407 | 4.08E-07 |
| CDH5 | −0.445 | 2.24E-08 |
| FBN1 | −0.468 | 3.29E-09 |
| EIF2B2 | −0.412 | 2.99E-07 |
| PLTP | −0.462 | 5.80E-09 |
| CDH11 | −0.512 | 5.66E-11 |
| CDH13 | −0.402 | 6.05E-07 |
| TNFAIP6 | −0.500 | 1.73E-10 |
| HSPG2 | −0.445 | 2.30E-08 |
| ITIH3 | −0.428 | 8.84E-08 |
| PCDH1 | −0.436 | 4.56E-08 |
| EFEMP1 | −0.533 | 6.03E-12 |
| DNAJC3 | 0.438 | 3.93E-08 |
| DSG2 | −0.473 | 2.13E-09 |
| EBI3 | −0.423 | 1.27E-07 |
| CYP1B1 | −0.429 | 8.02E-08 |
| SVEP1 | −0.505 | 1.04E-10 |
| ADAMTS13 | −0.441 | 3.24E-08 |
| ADAMTSL2 | −0.423 | 1.33E-07 |
| EVC2 | −0.452 | 1.27E-08 |
| PGLYRP2 | −0.532 | 6.47E-12 |
| MMP19 | −0.415 | 2.37E-07 |
| CHID1 | 0.414 | 2.53E-07 |
| CDHR5 | −0.431 | 6.94E-08 |
| PCDH12 | −0.476 | 1.71E-09 |
| FBLN5 | −0.481 | 1.03E-09 |
| NOTCH3 | −0.409 | 3.69E-07 |
| TFR2 | −0.421 | 1.46E-07 |
| RFNG | −0.476 | 1.68E-09 |

# Methods

### Reagents and tools table

| Reagent/resource | Reference or source | Identifier or catalog number |
|---|---|---|
| **Experimental models** | | |
| NA | | |
| **Recombinant DNA** | | |
| NA | | |
| **Antibodies** | | |
| Rabbit VE-cadherin | Abcam | Cat#ab313632 |
| Rabbit MMP2 | Proteintech | Cat#10373-2-AP |
| Mouse β-actin | Abcam | Cat#ab6276 |
| Goat anti-rabbit HRP IgG (H + L) | Servicebio | Cat# GB23303 |
| Goat anti-mouse HRP IgG (H + L) | Servicebio | Cat#GB23301 |
| **Oligonucleotides and other sequence-based reagents** | | |
| NA | | |
| **Chemicals, enzymes and other reagents** | | |
| Isovalerylcarnitine | TargetMol | Cat#T19384 |
| Calpeptin | Selleck | Cat#S7396 |
| Fetal Bovine Serum | BDBIO | Cat#F800-050 |
| RPMI 1640 | Gibco | Cat#12633020 |
| F-12K | Gibco | Cat#21127030 |
| Calpain Activity Assay Kit | Abcam | Cat#ab65308 |
| Protease Inhibitor Cocktail | Sigma-Aldrich | Cat#P8340 |
| Phenylmethylsulfonyl fluoride | Merck | Cat#329-98-6 |
| Matrigel | Acrobiosystems | Cat#AC-M082704 |
| Penicillin-Streptomycin | Solarbio | Cat#P1400 |
| Trypsin | EallBio | Cat#01.13008-25 |
| BCA Protein Assay Kit | CWBIO | Cat#CW0014S |
| 4% paraformaldehyde | Servicebio | Cat#G1101 |
| Skimmed milk powder | Servicebio | Cat#GC310001 |
| Crystal violet | Servicebio | Cat#G1014 |
| **Software** | | |
| Graphpad PRISM (version 10.1.1) | Graphpad | |
| R software (version 4.1.1) | Open access | |
| MetaboAnalyst 6.0 | Open access | |
| OriginPro (version 2021) | Open access | |
| Orange (version 3.36.1) | Open access | |
| Metascape (v3.5.20250701) | Zhou et al, 2019 | |
| **Other** | | |
| ChemiDoc™ Imaging System | BIO-Rad | |
| Model 550 Automatic Microplate Reader | BIO-Rad | |
| Leica DM 2000 light microscope | Leica Microsystems | |

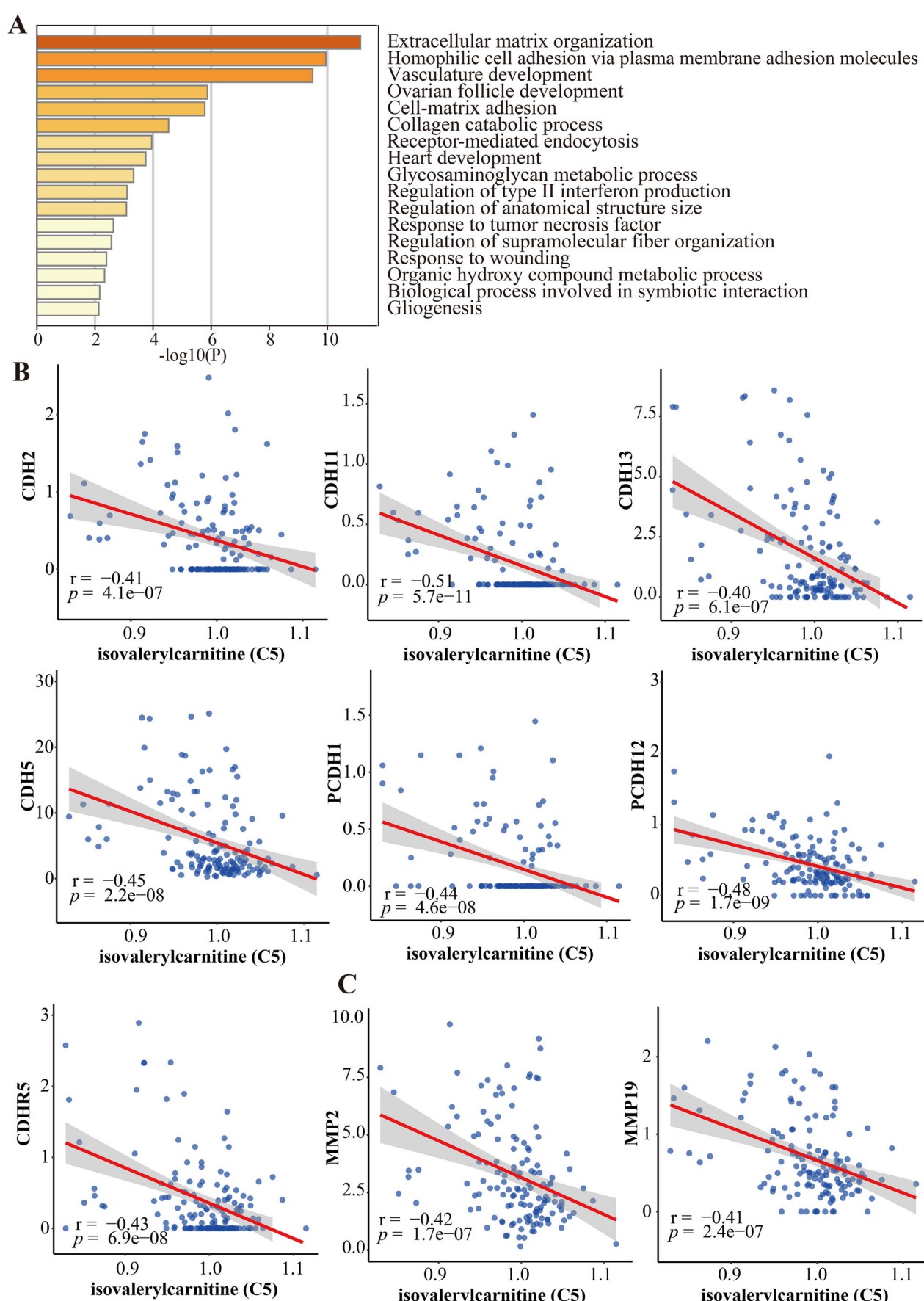

**Figure 5.   Isovalerylcarnitine (C5)-associated proteomic changes in gastric cancer.**

(**A**) Gene Ontology-Biological Process (GO-BP) enrichment analysis of GC protein biomarkers highly associated with isovalerylcarnitine (C5). Key biological processes include extracellular matrix organization, homophilic cell adhesion via plasma membrane adhesion molecules, and cell-matrix adhesion. Statistical analyses were performed using hypergeometric test. (**B**) Correlation between isovalerylcarnitine (C5) and cadherin family members (CDH2, CDH5, CDH11, CDH13, CDHR5, PCDH1, and PCDH12). Statistical analyses were performed using Pearson correlation test. (**C**) Correlation between isovalerylcarnitine (C5) and matrix metalloproteinase (MMP) family members (MMP2 and MMP19). Statistical analyses were performed using Pearson correlation test. Source data are available online for this figure.

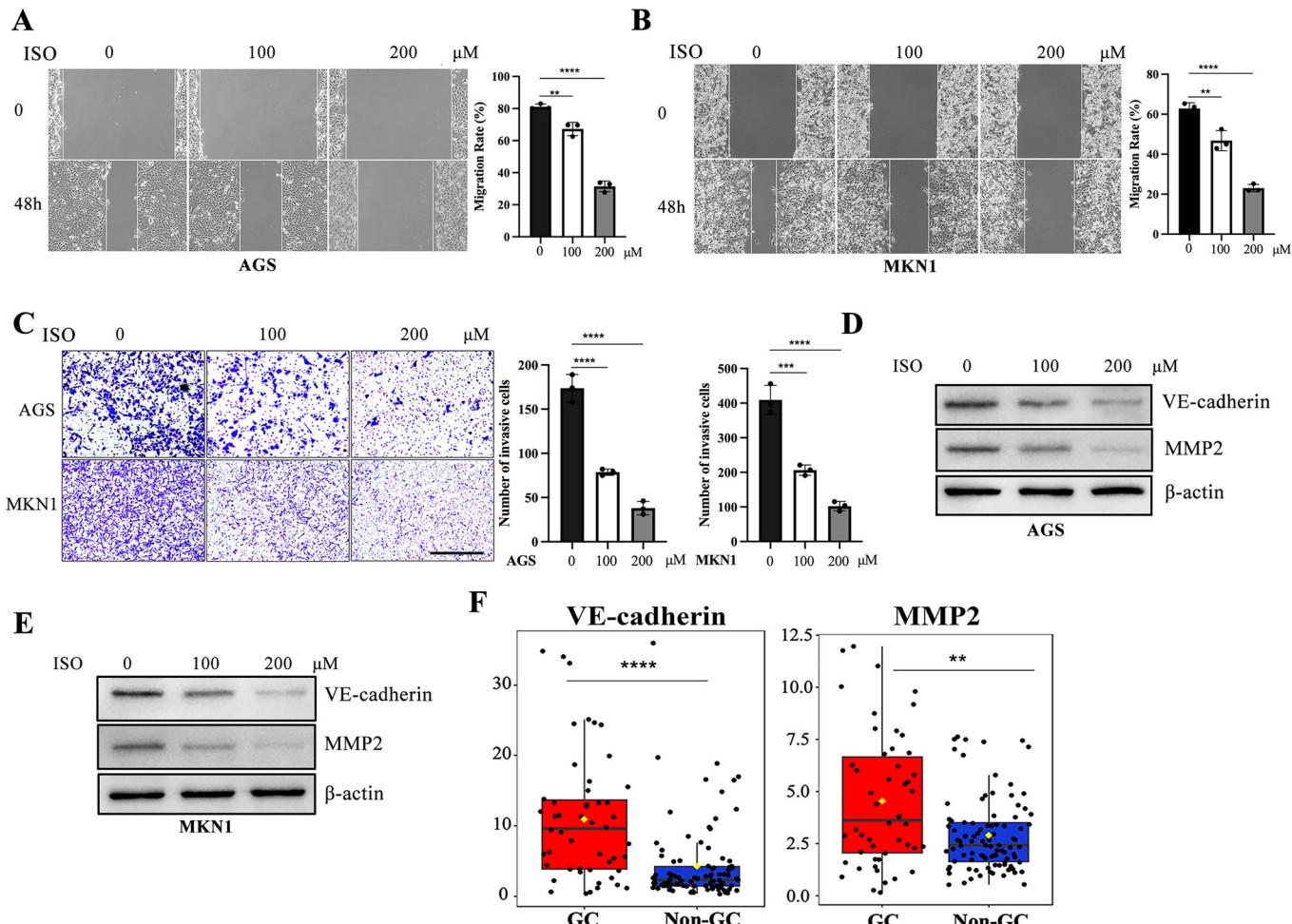

**Figure 6.   Isovalerylcarnitine (C5) suppresses gastric cancer metastasis.**

(**A**) Cell migration ability was estimated using scratch wound healing assays. Isovalerylcarnitine (C5) inhibited migration of AGS cells. Data were mean ± SD, $N = 3$ biological replicates, one-way ANOVA. The exact adjusted $P$ values: 0 μM vs 100 μM $p = 0.0031$; 0 μM vs 200 μM $P < 0.0001$. $^{**}P < 0.01$, $^{****}P < 0.0001$. ISO is short for isovalerylcarnitine (C5). (**B**) Cell migration ability was estimated using scratch wound healing assays. Isovalerylcarnitine (C5) inhibited migration of MKN1 cells. Data were mean ± SD, $N = 3$ biological replicates, one-way ANOVA. The exact adjusted $p$ values: 0 μM vs 100 μM $P = 0.0025$; 0 μM vs 200 μM $P < 0.0001$. $^{**}P < 0.01$, $^{****}P < 0.0001$. ISO is short for isovalerylcarnitine (C5). (**C**) Cell invasion ability was estimated using transwell assays. Isovalerylcarnitine (C5) inhibited invasion of AGS cells and MKN1 cells. Data were mean ± SD, $N = 3$ biological replicates, one-way ANOVA. The exact adjusted $P$ values in AGS: 0 μM vs 100 μM $P < 0.0001$; 0 μM vs 200 μM $P < 0.0001$. The exact adjusted $p$ values in MKN1: 0 μM vs 100 μM $P = 0.0002$; 0 μM vs 200 μM $p < 0.0001$. $^{***}P < 0.001$, $^{****}P < 0.0001$. Scale bars: 400 μm. ISO is short for isovalerylcarnitine (C5). (**D, E**) Western blot experiments showed that isovalerylcarnitine (C5) inhibited the expression of VE-cadherin and MMP2 in AGS cells (**D**) and MKN1 cells (**E**). ISO is short for isovalerylcarnitine (C5). (**F**) Differential expression of plasma VE-cadherin, and MMP2 between GC patients and non-GC controls. The exact adjusted $P$ values for VE-cadherin: $P < 0.0001$; for MMP2: $P = 0.0056$. $^{**}P < 0.01$, $^{****}P < 0.0001$. Box plots show the median (center line), the 25th and 75th percentiles (lower and upper bounds of the box), and whiskers extending up to 1.5 times the interquartile range from the box limits. Data points beyond this range are considered outliers and are shown individually. Source data are available online for this figure.

## Study design and participants

A total of 597 participants were enrolled, including 277 GC patients and 320 non-GC controls. Specifically, the discovery dataset comprised 144

individuals (50 GC, 94 non-GC) recruited at Zhejiang Cancer Hospital from January to December 2022; the validation dataset included 453 individuals (227 GC, 226 non-GC) enrolled between July 2010 and June 2021 across seven centers—Zhejiang Cancer Hospital, Sichuan Cancer

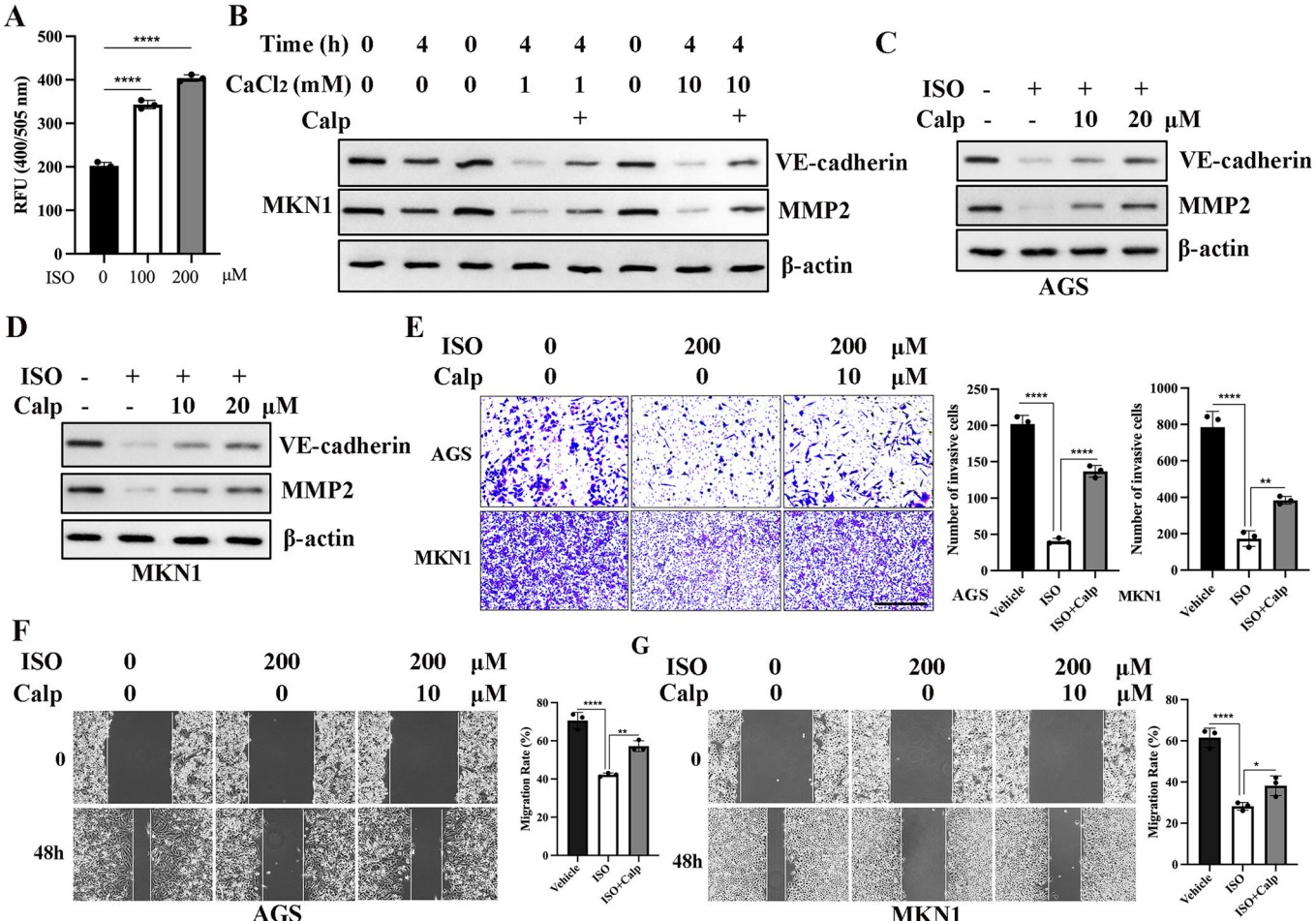

**Figure 7. Isovalerylcarnitine (C5) suppresses gastric cancer metastasis through calpain activation.**

(A) MKN1 cells treated with different concentrations of isovalerylcarnitine (C5) for 24 h were collected to detect calpain activity. Data were mean ± SD, $N = 3$ biological replicates, one-way ANOVA. The exact adjusted $P$ values: 0 μM vs 100 μM $P < 0.0001$; 0 μM vs 200 μM $P < 0.0001$. ****$P < 0.0001$. ISO is short for isovalerylcarnitine (C5). (B) MKN1 cells were collected and lysed, and the supernatant was incubated with or without CaCl$_2$ and 20 μM Calpeptin for 4 h. Then, western blotting was conducted to detect VE-cadherin and MMP2 expression. (C, D) Calpain inhibitor calpeptin reversed the inhibitory effects of isovalerylcarnitine (C5) (200 μM) on the expression of VE-cadherin and MMP2 in a dose-dependent manner in AGS cells (C) and MKN1 cells (D). ISO is short for isovalerylcarnitine (C5); Calp is short for Calpeptin. (E) Cell invasion ability was estimated using transwell assays. Calpeptin reversed the inhibitory effects of isovalerylcarnitine (C5) on the invasion of AGS cells and MKN1 cells. Data were mean ± SD, $N = 3$ biological replicates, one-way ANOVA. The exact adjusted $P$ values in AGS: Vehicle vs ISO $p < 0.0001$; ISO vs ISO+Calp $P < 0.0001$. The exact adjusted $P$ values in MKN1: Vehicle vs ISO $P < 0.0001$; ISO vs ISO+Calp $P = 0.0071$. **$P < 0.01$, ****$P < 0.0001$. Scale bars: 400 μm. ISO is short for isovalerylcarnitine (C5); Calp is short for Calpeptin. (F) Cell migration ability was estimated using scratch wound healing assays. Calpeptin reversed the inhibitory effects of isovalerylcarnitine (C5) on the migration of AGS cells. Data were mean ± SD, $N = 3$ biological replicates, one-way ANOVA. The exact adjusted $P$ values: Vehicle vs ISO $P < 0.0001$; ISO vs ISO+Calp $P = 0.0016$. **$P < 0.01$, ****$P < 0.0001$. ISO is short for isovalerylcarnitine (C5); Calp is short for Calpeptin. (G) Cell migration ability was estimated using scratch wound healing assays. Calpeptin reversed the inhibitory effects of isovalerylcarnitine (C5) on the migration of MKN1 cells. Data were mean ± SD, $N = 3$ biological replicates, one-way ANOVA. The exact adjusted $P$ values: Vehicle vs ISO $P < 0.0001$; ISO vs ISO+Calp $P = 0.0386$. *$P < 0.05$, ****$P < 0.0001$. ISO is short for isovalerylcarnitine (C5); Calp is short for Calpeptin. Source data are available online for this figure.

Hospital, Zhejiang Provincial Hospital of Chinese Medicine, The First People's Hospital of Daishan, The People's Hospital of Fenghua, The People's Hospital of Xinchang, and Tiantai Chinese Medicine Hospital. This study adhered strictly to the principles of the Declaration of Helsinki and the Department of Health and Human Services Belmont Report, with the approval by the ethics committee of Zhejiang Cancer Hospital (approval No. IRB-2022-271). All patients provided written informed consent before participating in the study.

GC patients were eligible if they (1) had histopathologically confirmed gastric cancer, (2) provided a preoperative plasma sample, (3) had not received neoadjuvant therapy, and (4) had complete clinical records and

follow-up data. Exclusion criteria were lack of a preoperative plasma sample, prior tumor treatment, or incomplete medical records. Tumor characteristics (pathology, size, location, differentiation) were obtained from medical records, and Tumor, Node, Metastasis (TNM) staging was assigned according to the American Joint Committee on Cancer (AJCC) 8th edition. Non-GC controls were collected from routine physical examinations or screening programs, including healthy, gastritis and intestinal metaplasia conditions. They provided a fasting plasma sample before endoscopy and had no history of cancer treatment. Controls using medications known to affect hematologic parameters (e.g., adrenocortical hormones and glucocorticoids) were excluded.

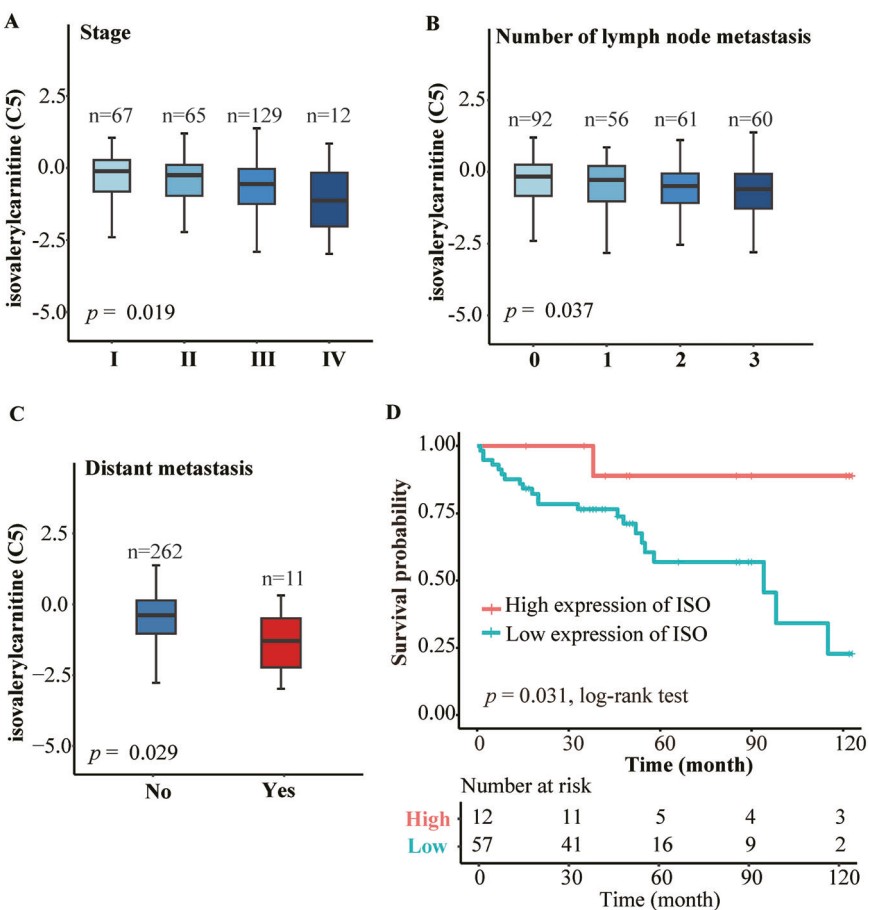

**Figure 8. Isovalerylcarnitine (C5) is negatively correlated with metastasis of gastric cancer patients.**

(A) Expression of isovalerylcarnitine (C5) with tumor stages of GC (sample number, $N = 273$), showing variations in expression levels. Kruskal–Wallis test, $P = 0.019$. Box plots show the median (centre line), interquartile range (bounds of box, 25th–75th percentile), and whiskers extending to the minimum and maximum values. (B) Expression of isovalerylcarnitine (C5) with the number of lymph node metastases in GC patients (sample number, $N = 269$), indicating a potential role in metastasis. Kruskal–Wallis test, $P = 0.037$. Box plots show the median (centre line), interquartile range (bounds of box, 25th–75th percentile), and whiskers extending to the minimum and maximum values. (C) Expression of isovalerylcarnitine (C5) with distant metastasis in GC patients (sample number, $N = 273$), further supporting its involvement in cancer metastasis. Wilcoxon rank-sum test, $P = 0.029$. Box plots show the median (centre line), interquartile range (bounds of box, 25th–75th percentile), and whiskers extending to the minimum and maximum values. (D) Kaplan–Meier curves of overall survival of patients in low (blue line) or high (red line) expression of isovalerylcarnitine (C5) with $P = 0.031$ by log-rank test, suggesting that low expression of isovalerylcarnitine (C5) may be associated with poorer prognosis. ISO is short for isovalerylcarnitine (C5). Source data are available online for this figure.

## Data collection

Before the gastroscopy procedure, comprehensive demographic and clinical information was collected, including age, sex, smoking history, drinking history, *Helicobacter pylori (H. pylori)* infection status, and traditional blood tumor biomarkers. The biomarkers, including carbohydrate antigen 724 (CA724), CA199, CA242, CA125, carcinoembryonic antigen (CEA), and alpha-fetoprotein (AFP), were analyzed following clinical instructions.

## Sample collection and processing

Peripheral blood (5 mL) was drawn from fasting participants before gastroscopy or surgery into ethylenediaminetetraacetic acid (EDTA) tubes, delivered on ice, and processed within one hour at 4 °C. Samples were centrifuged at 1800× *g* for 10 min at 4 °C, and the plasma layer was aliquoted and stored at –80 °C according to a uniform SOP (GB/T

38576-2020) across all centers. Concurrently, paired tumor and adjacent non-tumor tissues were collected from GC patients in the discovery dataset during surgical resection. Non-tumor adjacent tissues (NATs) were sampled ~2 cm from the tumor margin; each specimen measured roughly 0.5 × 0.5 cm, and typically 4–5 matched tumor–NAT pairs were obtained per patient. All tissues underwent cold ischemia for under 30 min before being snap-frozen at –80 °C. A board-certified pathologist confirmed that tumor sections contained ≥60% tumor nuclei and < 20% necrosis by examining the top and bottom of each specimen. The pathologists were blinded to any information about metabolome analysis.

## Metabolome analysis in the discovery dataset

Blood and tissue samples from the discovery dataset were sent to CALIBRA Corporation's affiliated lab, Calibra Lab at DIAN Diagnostics in Hangzhou, China, for untargeted metabolomic

profiling using the CalOmics platform. The plasma and tissue samples were prepared randomly to minimize subjective bias. Details of the method can be found in a prior study (Shen et al, 2020). Tissues were minced and homogenized prior to extraction, and plasma and tissue aliquots were treated identically: each 50 μL sample was mixed with 200 μL methanol (1:4 v/v), shaken, and centrifuged at 4000× $g$ for 10 min at 20 °C. Four 100 μL aliquots of supernatant were transferred to 96-well plates, dried under nitrogen for ≥3 h, then reconstituted for ultra performance liquid chromatography tandem mass spectrometry (UPLC-MS/MS) analysis. Chromatography was performed on a Waters ACQUITY 2D UPLC system coupled to a Q Exactive Plus Orbitrap (Thermo Fisher) at 35,000 resolutions, scanning $m/z$ 70–1000. Four complementary methods were applied: (1) positive electrospray Ionization (ESI) on a BEH C18 column (2.1 × 100 mm, 1.7 μm; Waters); the mobile solutions used in the gradient elution were water and methanol containing 0.05% perfluoropentanoic acid (PFPA) and 0.1% formic acid (FA); (2) negative ESI on BEH C18 with water/methanol (6.5 mM ammonium bicarbonate, pH 8); (3) positive ESI on BEH C18 with water/methanol/acetonitrile (0.05% PFPA, 0.01% FA); and (4) negative ESI on a BEH Amide HILIC column (2.1 × 150 mm, 1.7 μm; Waters) using water/acetonitrile (10 mM FA). The QE was operated in negative ESI mode with alternating MS and data-dependent MS2 scans using dynamic exclusion. The scan range was 70–1000 $m/z$. Capillary temperature was 350 °C, with sheath and aux gas flow rates of 40 and 5, respectively, for both ion modes.

After raw data preprocessing, including peak detection, alignment, and annotation using in-house software, metabolites were identified by matching against a custom spectral library containing over 3300 purified standards analyzed on the same experimental platforms. Identification required satisfying three stringent criteria: (1) a narrow retention index (RI) window, (2) accurate mass within 10 ppm deviation, and (3) high forward and reverse MS/MS spectral similarity scores. These criteria enabled reliable distinction of nearly all isomers. Multiple quality control (QC) samples were incorporated, including pooled samples (comprising aliquots of all experimental samples) analyzed repeatedly to assess technical variation (16 plasma technical replicates and 12 tissue technical replicates in the study), extracted water blanks, and commercial plasma to monitor instrument performance. A mixture of internal standards was added to each sample for chromatographic alignment and instrument stability assessment. Instrumental variability was evaluated by calculating the median relative standard deviation (RSD) of internal standards, while overall experimental variability was assessed by the median RSD of endogenous metabolites in pooled QC samples (RSD ≤ 5%).

## Metabolome analysis in the validation dataset

Plasma samples in the validation dataset were processed and analyzed by UPLC-MS at the Instrumental Analysis Cancer Facility, Shanghai Jiao Tong University. Details of the method have been described in previous publications (Xu et al, 2023; Liu et al, 2024). For each sample, 50 μL of plasma was mixed with 200 μL of methanol:acetonitrile (1:1, v/v), incubated at –20 °C for at least 2 h, and centrifuged at 13,202× $g$ for 20 min. The supernatant was vacuum-dried for ≥3 h, reconstituted in 100 μL methanol:water (3:7, v/v), vortexed for 3 min, and centrifuged again at 13,202× $g$

and 4 °C for 15 min to remove particulates. QC samples were generated by pooling 140 aliquots (70 GC patients and 70 non-GC controls), divided into three aliquots, and stored at –80 °C. Chromatographic separation was performed on a Thermo Scientific Vanquish UHPLC system coupled to a Q Exactive Plus mass spectrometer (Thermo Scientific, USA), using an ACQUITY UPLC HSS T3 column (100 × 2.1 mm, 1.7 μm; Waters). Mobile phase A consisted of water with 0.1% formic acid, and mobile phase B was acetonitrile with 0.1% formic acid. The gradient ran from 99% A/1% B to 0% A/100% B over 12 min, held for 1 min, at a flow rate of 0.4 mL/min and column temperature of 40 °C. A 1 μL aliquot of each sample was injected. The Q Exactive Plus operated in data-dependent acquisition (DDA) mode in both positive and negative HESI, scanning $m/z$ 67–1000 with a full-scan resolution of 70,000. Spray voltage was set to 3.2 kV (positive) or 2.8 kV (negative), capillary temperature to 320 °C, and S-lens RF level to 50 V. Raw data were processed in Progenesis QI (v2.3; Waters, UK) for peak picking, alignment, and normalization. The metabolites were identified through the molecular formula search against the Human Metabolome Database (HMDB, https://hmdb.ca/) (Wishart et al, 2022).

## Protein profiling in the discovery dataset

Protein profiling was performed as previously described (Shi et al, 2023), using Orbitrap-based mass spectrometers. Peptides were analyzed on an Orbitrap Fusion system operated in data-dependent acquisition (DDA) mode. Dried peptide samples were reconstituted in Solvent A (0.1% formic acid in water) and loaded onto a homemade trap column (max pressure: 280 bar), followed by separation on a homemade silica microcolumn. Elution was performed with a gradient of mobile phase B (0.1% formic acid in acetonitrile) at 350 nL/min over 75 min. Peptides were ionized at 2 kV and analyzed over an $m/z$ range of 300–1400. Full MS scans were acquired at a resolution of 120,000 (Orbitrap Fusion) or 60,000 (Q Exactive HF), followed by up to 20 MS/MS scans per cycle. Raw MS data were processed using the Firmiana platform and searched against the human NCBI RefSeq database via Mascot 2.3. Protein identification was based on a 0.5% false discovery rate. Data were normalized using intensity-based absolute quantification and fraction of total methods, with missing values imputed by the minimum observed value. The platform exhibited high reproducibility, with Spearman correlation coefficients >0.9 across standards.

## Model construction based on machine learning algorithms

For diagnostic model building, machine learning algorithms(NN, SVM, RR, LR and NB) were applied utilizing a dedicated data mining toolkit, Orange (version 3.36.1, the Bioinformatics Lab at the University of Ljubljana, Slovenia) (Demšar et al, 2013). To ensure reproducibility, the full Orange workflow file (.ows), classifier parameter settings, and input feature details have been made publicly available at GitHub (https://github.com/Liu-Memory/orange_machine_learning). For model-based feature selection (lasso regression score [LRScore]-based method), the 'Rank' widget in the toolbox was used. This widget was connected to the lasso regression model to obtain the LRScore for each feature. For model building, stratified fivefold cross-validation was used to train neural network,

support vector machine, ridge regression, lasso regression, and naive Bayes in the discovery dataset. To validate their robustness and generalizability, the constructed models were subsequently subjected to testing within a validation dataset. Each of the integrated machine learning models generates an output probability score for cancer presence in patients, spanning the continuum from 0 to 1.

## Bidirectional Mendelian randomization analysis

Data sources were listed in Appendix Table S9. Summary-level statistics of genetic associations among 1400 plasma metabolites were extracted from a comprehensive metabolomic genome-wide association study (GWAS) of 8299 European ancestry participants (GWAS Catalog accession GCST90199621–GCST90201020) (Chen et al, 2023). GC association data were obtained from FinnGen GWAS comprising 1307 cases and 287,137 controls. The association analyses in the GWAS summary data were performed with adjustments for sex, age, and the first 10 principal components. For each metabolite, single nucleotide polymorphisms (SNPs) with $P < 1 \times 10^{-5}$ and minor allele frequency ≥0.01 were retained and pruned by linkage disequilibrium ($r^2 < 0.01$, 500 kb) to serve as instrumental variables (IVs). In reverse MR, we selected GC-associated IVs at $P$ value $< 5 \times 10^{-6}$. Causal estimates were derived primarily by the inverse-variance weighted method, combining SNP–exposure and SNP–outcome β coefficients and standard errors (Palmer et al, 2011). We conducted sensitivity analyses using simple-median and weighted-median estimators to account for potential invalid IVs (Burgess et al, 2017), a likelihood-based approach assuming a linear exposure-outcome relationship, and MR-Egger regression to detect directional pleiotropy (intercept $P < 0.05$) (Bowden et al, 2015). To correct for multiple testing, the Bonferroni threshold was set at $P < 3.57 \times 10^{-5}$ (0.05/1399), with $3.57 \times 10^{-5} < P < 0.05$ considered suggestive (Larsson et al, 2017). All MR analyses were performed in R (v4.2.2) using the "TwoSampleMR," "Mendelian-Randomization," and "MRPRESSO" packages.

## Western blots

MKN1 and AGS cells were harvested and lysed in RIPA buffer (50 mM Tris-HCl pH 7.4, 1% NP40, 0.5% Na-deoxycholate, 0.1% SDS, 150 mM NaCl, 2 mM EDTA, and 50 mM NaF) supplemented with protease inhibitor cocktail (Sigma) and Phenylmethylsulfonyl Fluoride (Sigma) on ice for 30 min. Cell lysates were cleared by centrifuging at 12,000 rpm, 4 °C for 20 min, and the protein concentration was determined by Bradford assay (Bio-Rad). Equal amounts of protein (20–40 µg) per sample were for the following IB. The antibodies used were anti-VE-cadherin (Abcam, ab313632, 1:1000 dilution), anti-MMP2 (Proteintech, 10373-2-AP, 1:1000 dilution), and anti-β-actin (Abcam, ab6276, 1:10,000 dilution).

## Calpain activity

Calpain activity assay was conducted in strict accordance with the protocol provided in the product manual (ab65308, Abcam). Briefly, MKN1 cells treated with vehicle or isovalerylcarnitine (C5) for 24 h, were collected, pelleted by centrifugation, followed by being resuspended in 100 µl extraction buffer and incubated on ice for 20 min. The tubes were tapped multiple times during incubation to achieve gentle mixing. After centrifugation with $10,000 \times g$ for

2 min, the cell lysate was diluted in 85 µl extraction buffer and transferred into 96-well plate. All inputs were standardized to same protein amount according to total protein measurement. 10 µl 10X reaction buffer and 5 µl calpain substrate (Ac-LLY-AFC) were added to each assay well and incubated at 37 °C for 1 h in dark. All samples were analyzed with a multilabel reader and expressed as relative fluorescent units (RFU).

## In vitro calpain assay

The MKN1 cells were lysed in buffer (50 mM Tris-HCl pH 7.4, 1% NP40, 0.5% Na-deoxycholate, 150 mM NaCl, and 50 mM NaF) containing protease inhibitors but no leupeptin. The extraction was centrifuged at 1000 g for 10 min at 4 °C, and the resulting supernatant was collected. The supernatant was then incubated at 30 °C with CaCl$_2$ and calpeptin for 4 h. The 5X loading buffer was added to stop the reaction, and the sample was boiled at 100 °C for 10 min. Finally, Western blot analysis was conducted to detect the expression levels of VE-cadherin and MMP2.

## Cell migration and invasion

For cell migration assay, MKN1 and AGS cells were trypsinized, counted, and seeded into six-well plates to reach near-confluence (~24 h). Using a sterile 10 µL pipette tip, a uniform scratch was made across the monolayer. After gently washing with serum-free medium to remove debris, images of wound closure were captured at 0 h and 48 h using an inverted microscope (×10). For cell invasion assay, transwell inserts (8 µm pore size) were precoated with 10% Matrigel to mimic the extracellular matrix. MKN1 and AGS cells were serum-starved, resuspended in serum-free medium, and seeded into the upper chamber, while the lower chamber contained complete medium with 10% FBS as a chemoattractant. After 24 h at 37 °C, non-invading cells on the upper membrane surface were removed with a cotton swab. Invaded cells on the lower surface were fixed in 4% paraformaldehyde for 15 min, stained with 0.1% crystal violet for 20 min, rinsed with PBS, and air-dried. Stained cells were then visualized under a microscope, and five random fields per insert were counted to quantify invasion.

## Statistical analysis

Sample size was chosen based on the need for statistical power. Power analysis was conducted on MetaboAnalyst 6.0 (https://www.metaboanalyst.ca/) to determine the sample size required for statistically significant machine learning (Pang et al, 2024). Specifically, the metabolites of 6 plasma samples (3/3, GC/Non-GC) were uploaded, and the predicted power was computed at a false discovery rate of 0.1. Features with >10% missing values were excluded, and remaining missing values were imputed at the limit of detection (1/5 of the minimum positive value of each variable) (Pang et al, 2024; Li et al, 2024a; Liu et al, 2025). To assess the impact of this imputation strategy on model performance, we conducted a sensitivity analysis using four alternative imputation methods: mean imputation, median imputation, k-nearest neighbor (KNN) imputation, probabilistic principal component analysis (PCA) imputation, confirming the stability of our findings across different imputation strategies (Appendix Table S14). The

**The paper explained**

**Problem**

Early detection of gastric cancer (GC) remains challenging, as conventional protein-based biomarkers (e.g., CEA, CA72-4) exhibit low sensitivity and specificity, particularly in early-stage disease, underscoring the need for more reliable noninvasive diagnostic tools. Notably, a recent large multi-center study by Chen et al demonstrated the diagnostic potential of metabolomics in GC using an external cohort. However, many earlier studies were limited by small sample sizes or single-center designs, which may constrain the generalizability of their findings. Moreover, although certain metabolic biomarkers have been implicated in GC risk, their functional roles in tumor progression and the underlying molecular mechanisms remain poorly understood. This knowledge gap has hindered the translation of metabolic discoveries into clinically actionable biomarkers or therapeutic targets for GC.

**Results**

This study advances the field by integrating large-scale metabolomics data from multi-center cohorts with machine learning to develop a robust diagnostic model for GC. We identified and validated a panel of six plasma metabolites, achieving superior diagnostic performance compared to traditional biomarkers. Notably, we demonstrated the consistent downregulation of isovalerylcarnitine (C5) in GC tissues and plasma, supported by Mendelian randomization, suggesting its causal role in GC risk. Mechanistically, we revealed that isovalerylcarnitine (C5) inhibits GC cell migration and invasion by activating calpain, leading to the cleavage of VE-cadherin and MMP2. Our findings provide both a high-performance diagnostic tool and novel insights into the metabolic mechanisms driving GC progression.

**Impact**

The evidence from this study underscores the transformative potential of metabolomics and machine learning in GC diagnosis and mechanistic research. Our diagnostic model, validated across diverse cohorts, offers a noninvasive, cost-effective alternative to traditional methods, particularly beneficial for early detection in high-risk populations. The discovery of isovalerylcarnitine (C5)'s tumor-suppressive role opens new avenues for targeted therapies aimed at modulating metabolic pathways in GC. Future research should focus on validating these findings in broader populations and exploring the clinical utility of metabolic biomarkers in precision medicine strategies for GC management.

discovery cohort was analyzed in a single batch using an LC-MS platform at Calibra, and the external validation cohort was analyzed in a single batch using an independent LC-MS platform at Shanghai Jiao Tong University. Within each cohort, raw metabolite intensities were normalized by the sum of peak areas, $\log_2$-transformed, and auto-scaled (mean-centering and division by each variable's standard deviation) to reduce intra-cohort variability. No datasets from different platforms were merged. Instead, differential metabolites were identified in the discovery set and subsequently tested in the independent validation set to evaluate cross-cohort generalizability. FCs were calculated as the mean ratio between groups, with significant up- or downregulation defined by FC > 1.50 or <0.67, respectively, and metabolites passing a false discovery rate < 0.05 were retained. PLS-DA was performed in MetaboAnalyst 6.0, and logistic regression evaluated associations with gastric cancer, yielding sensitivity, specificity, accuracy, and AUC metrics. AUCs (95% CI) were computed using the receiver operating characteristic (ROC) Analysis widget in Orange (version

3.36.1, the Bioinformatics Lab at the University of Ljubljana, Slovenia) and OriginPro (version 2021, OriginLab in Northampton, Massachusetts, USA). Additional statistical tests—including Student's $t$ tests, chi-square tests, heatmaps ('pheatmap'), forest plots ('forestplot'), boxplots ('geom_boxplot'/'stat_boxplot'), and confusion matrices ('confusionMatrix')—were conducted in R 4.1.1. For pathway analysis, quantitative metabolite enrichment was performed in MetaboAnalyst 6.0 against the KEGG database, and signature proteins underwent Gene Ontology and KEGG enrichment via Metascape (https://metascape.org/)(Zhou et al, 2019).

## Data availability

The metabolomics data have been deposited to MetaboLights (Yurekten et al, 2024) repository with the study identifier MTBLS12684 (https://www.ebi.ac.uk/metabolights/reviewerbe3ef273-b05d-4cb9-9262-d220d55ccf2b). To enhance the reproducibility of our study, we have uploaded the complete model construction and validation workflow, the structure and parameters of classifiers (file name: Machine Learning_training and test.ows) to the GitHub (https://github.com/Liu-Memory/orange_machine_learning).

The source data of this paper are collected in the following database record: biostudies:S-SCDT-10_1038-S44321-025-00325-0.

## Peer review information

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

## Acknowledgements

This work was supported by the National Natural Science Foundation of China (82504487; 823B2050) and the Natural Science Foundation of Zhejiang Province (LQN25H260001) and the Medical and Health Research Project of Zhejiang Province (2024KY050; 2025KY043). We thank the hospitals and staff for their cooperation in providing, sorting, and verifying the data. We thank all the participants for donating their bio-samples and clinical information to this study.

## Author contributions

**Juan Zhu**: Conceptualization; Data curation; Formal analysis; Methodology; Writing—original draft. **Yida Huang**: Conceptualization; Data curation; Validation; Methodology; Writing—review and editing. **Xue Li**: Conceptualization; Methodology; Writing—review and editing. **Bin Liu**: Validation; Methodology; Writing—review and editing. **Li Yuan**: Data curation; Writing—review and editing. **Le Wang**: Investigation; Writing—review and editing. **Kun Qian**: Methodology; Writing—review and editing. **Yingying Mao**: Methodology; Writing—review and editing. **Yongjie Xu**: Conceptualization; Data curation; Formal analysis; Validation; Methodology; Writing—review and editing. **Lingbin Du**: Supervision; Writing—review and editing. **Xiangdong Cheng**: Supervision; Writing—review and editing.

Source data underlying figure panels in this paper may have individual authorship assigned. Where available, figure panel/source data authorship is listed in the following database record: biostudies:S-SCDT-10_1038-S44321-025-00325-0.

## Disclosure and competing interests statement

The authors declare the following competing interests. The authors have filed patents using the methods and technologies to analyze metabolites.

