## [Peer Review File · EMBO Molecular Medicine]

Metabolic Signatures for Gastric Cancer Diagnosis and Mechanistic Insights: A Multicenter Study

Juan Zhu, Yida Huang, Xue Li, Bin Liu, Li Yuan, Le Wang, Kun Qian, Yingying Mao, Yongjie Xu, Lingbin Du, and Xiangdong Cheng

Corresponding authors: Lingbin Du (dulb@zjcc.org.cn) , Yongjie Xu (Jayshu@bjmu.edu.cn), Xiangdong Cheng (chengxd@zjcc.org.cn)

Review Timeline:

Submission Date:	21st May 25
Editorial Decision:	12th Jun 25
Revision Received:	13th Aug 25
Editorial Decision:	28th Aug 25
Revision Received:	3rd Sep 25
Editorial Decision:	29th Sep 25
Revision Received:	2nd Oct 25
Accepted:	9th Oct 25

Editor: Zeljko Durdevic

Transaction Report:

12th Jun 2025

Dear Prof. Du,

Thank you for the submission of your manuscript to EMBO Molecular Medicine. We have now received feedback from the two reviewers who agreed to evaluate your manuscript. As you will see from the reports, both referees recognize potential interest of the study, but they also raise serious and partially overlapping concerns that should be addressed in a major revision. If you would like to discuss further the points raised by the referees, I am available to do so via email or video. Let me know if you are interested in this option.

We would welcome the submission of a revised version within three months for further consideration. Please let us know if you require longer to complete the revision.

I look forward to receiving your revised manuscript.

Yours sincerely,

Zeljko Durdevic

Zeljko Durdevic
Senior Editor
EMBO Molecular Medicine

We require:

- 1) A .docx formatted version of the manuscript text (including legends for main figures, EV figures and tables). Please make sure that the changes are highlighted to be clearly visible.
- 2) Individual production quality figure files as .eps, .tif, .jpg (one file per figure). For guidance, download the 'Figure Guide PDF': (<https://www.embopress.org/page/journal/17574684/authorguide#figureformat>).
- 3) A .docx formatted letter INCLUDING the reviewers' reports and your detailed point-by-point responses to their comments. As part of the EMBO Press transparent editorial process, the point-by-point response is part of the Review Process File (RPF), which will be published alongside your paper.
- 4) A complete author checklist, which you can download from our author guidelines (<https://www.embopress.org/page/journal/17574684/authorguide#submissionofrevisions>). Please insert information in the checklist that is also reflected in the manuscript. The completed author checklist will also be part of the RPF.
- 5) Please note that all corresponding authors are required to supply an ORCID ID for their name upon submission of a revised manuscript.
- 6) It is mandatory to include a 'Data Availability' section after the Materials and Methods. Before submitting your revision, primary

datasets produced in this study need to be deposited in an appropriate public database, and the accession numbers and database listed under 'Data Availability'. Please remember to provide a reviewer password if the datasets are not yet public (see <https://www.embopress.org/page/journal/17574684/authorguide#dataavailability>).

12) Author contributions: You will be asked to provide CRediT (Contributor Role Taxonomy) terms in the submission system. These replace a narrative author contribution section in the manuscript.

13) A Conflict of Interest statement should be provided in the main text.

14) Every published paper now includes a 'Synopsis' to further enhance discoverability. Synopses are displayed on the journal webpage and are freely accessible to all readers. They include a short stand first (maximum of 300 characters, including space) as well as 2-5 one-sentences bullet points that summarizes the paper. Please write the bullet points to summarize the key NEW findings. They should be designed to be complementary to the abstract - i.e. not repeat the same text. We encourage inclusion of key acronyms and quantitative information (maximum of 30 words / bullet point). Please use the passive voice. Please attach

these in a separate file or send them by email, we will incorporate them accordingly.

15) Include a Reagents and Tools Table as part of the Methods section, which can be downloaded from our author guidelines (<https://www.embopress.org/page/journal/17574684/authorguide#structuredmethods>)

***** Reviewer's comments *****

Referee #1 (Remarks for Author):

General Comments

This manuscript reports a multi-center study integrating untargeted metabolomics and machine learning to develop a six-metabolite diagnostic panel for gastric cancer (GC), alongside an exploratory investigation of the role of isovalerylcarnitine (C5) in tumor invasion. The incorporation of a multi-center cohort and the integration of metabolomic and proteomic analyses enhance the translational relevance of the study. However, several critical limitations reduce the overall strength and impact of the work. These include the absence of benchmarking against recent state-of-the-art metabolomics-based diagnostic models, insufficient methodological transparency in model development, suboptimal figure presentation, and incomplete validation of the proposed mechanistic pathway. Collectively, these issues hinder the interpretability, reproducibility, and translational potential of the findings.

In its current form, the manuscript does not meet the standards expected of a high-impact journal. Nevertheless, with comprehensive revision and additional validation, the study could become a meaningful contribution to the field of metabolomics-driven cancer diagnostics.

Major Concerns

1. Insufficient Benchmarking Against State-of-the-Art Metabolomics-Based Diagnostic Models

While the Met-NN(6) model is reported to outperform conventional protein biomarkers (e.g., CA724, CEA), it is not evaluated against other contemporary metabolomics-based classifiers. Recent studies-such as Chen et al. (Nat Commun, 2024) and Huang et al. (JAMA Netw Open, 2021)-have developed high-performing GC predictors using machine learning or ensemble frameworks in large cohorts. Without such benchmarking, the added diagnostic value of the proposed six-metabolite panel remains unclear. A comparative evaluation with these established models would substantively strengthen the manuscript's claims of clinical relevance and novelty.

2. Lack of Methodological Transparency in Model Training and Evaluation

The manuscript omits key details regarding the development and evaluation of the machine learning models, including:

The structure of the classifiers (e.g., neural network architecture, number of hidden layers, activation functions);

Hyperparameters (e.g., learning rate, batch size, dropout rates, regularization strategies);

Reproducibility controls (e.g., random seed initialization, software versions, number of independent training runs).

This lack of detail limits reproducibility and raises concerns regarding potential overfitting. The authors should fully disclose the model architecture, training pipeline, optimization procedures, and cross-validation strategy. Provision of open-source code and/or trained model weights would further enhance transparency and foster reproducibility.

3. Inconsistent and Confusing Presentation of Figure 2

Several issues in Figure 2 impede interpretation:

In Figure 2B, the use of "unsig" as a label is unconventional and should be replaced with "not significant."

The heatmap color scale labeled as "FoldChange" appears to represent log₂-transformed data. This should be clearly corrected and relabeled (e.g., "log₂(FC, GC/non-GC)").

Figure 2E presents the classification performance of a 26-metabolite panel across five machine learning algorithms. However, this analysis is not sufficiently described in the main text (Page 5, Line 163), and its visual similarity to Figure 3A (which pertains to the six-metabolite model) creates confusion.

The authors should revise both the text and figure legends for clarity. It may also be beneficial to relocate Figure 2E to the modeling results section alongside Figure 3A for improved logical flow.

4. Mechanistic Interpretation of the C5-Calpain Axis Requires Further Validation

The proposed mechanism in which isovalerylcarnitine (C5) suppresses GC cell migration via calpain-mediated degradation of VE-cadherin and MMP2 remains insufficiently substantiated. While the hypothesis is biologically plausible, the current evidence is primarily correlative. Notably:

No direct evidence of calpain activation following C5 treatment is provided;

Calpain activity or substrate cleavage is not validated (e.g., through activity assays or co-immunoprecipitation);

Alternative regulatory pathways are not explored or excluded.

To support the mechanistic claims, additional experiments should be conducted, or at minimum, the interpretation should be reframed as preliminary and exploratory.

Conclusion

The manuscript addresses a clinically significant challenge and introduces a potentially useful diagnostic approach leveraging integrated metabolomics and machine learning. However, substantial revisions are required to improve methodological rigor, clarify data presentation, conduct appropriate benchmarking, and provide stronger mechanistic support. Should these issues be adequately addressed, the study could provide a meaningful advancement in the development of metabolomics-based diagnostics for gastric cancer.

Referee #2 (Remarks for Author):

This multicenter study integrates untargeted metabolomics and machine learning to develop a plasma metabolite panel for gastric cancer (GC) diagnosis and mechanistic investigation. Using UPLC-MS/MS analysis of 597 participants, the authors identified a 6-metabolite diagnostic panel (including isovalerylcarnitine/C5) that outperforms conventional biomarkers (AUC: 0.920-0.982). They further validated C5 as a tumor-suppressive metabolite through Mendelian randomization and functional assays, demonstrating its role in inhibiting GC metastasis via calpain-mediated cleavage of VE-cadherin and MMP2. The study provides valuable insights into metabolic reprogramming in GC and offers a promising non-invasive diagnostic tool.

Major Concerns and Revisions Required

1. Sample Diversity and Generalizability

Issue: The cohort is exclusively Chinese. Given GC's global heterogeneity (e.g., *H. pylori* strain variations, dietary influences), results may not generalize.

Revision: Discuss limitations regarding ethnic/geographic generalizability in the Discussion. If feasible, validate the model in a non-Asian cohort (even preliminarily).

2. Control Group Composition

Issue: Non-GC controls included healthy individuals and gastritis patients but excluded high-risk premalignant conditions (e.g., intestinal metaplasia).

Revision:

Add analysis comparing metabolite levels between GC vs. premalignant groups (if biobanked samples exist).

Explicitly state this limitation and propose future validation in high-risk cohorts.

3. Mechanistic Ambiguity in C5 Signaling

Issue: The exact molecular link between C5-activated calpain and downregulation of VE-cadherin/MMP2 remains incompletely resolved (Section 6, Pg 10). Proposed mechanisms (direct degradation vs. calcium depletion) lack experimental validation.

Revision:

Include calpain activity assays after C5 treatment to confirm activation.

Test intracellular calcium flux in response to C5 using fluorescent probes (e.g., Fluo-4).

Discuss alternative pathways (e.g., mTOR/AMPK) given C5's role in leucine metabolism.

4. Technical and Analytical Details

Issue:

Missing value handling ("imputed at the limit of detection") lacks justification (Methods, Pg 15).

Tissue metabolomics validation (n=50) is underpowered for subgroup analyses.

Revision:

Clarify imputation methods and sensitivity analysis (e.g., impact on model AUC).

Acknowledge limited statistical power for tissue-level correlations.

5. Data Presentation

Issue:

Fig 7D suggests lower C5 correlates with worse survival, but the figure legend states "high expression of C5 may be associated with poorer prognosis" (contradicting results).

Table 1: Validation cohort has significant age/sex imbalances ($P < 0.001$ for age distribution).

Revision:

Correct Fig 7D legend and Kaplan-Meier labels.

Address cohort imbalances via stratified analysis or covariate adjustment.

Minor Revisions

Abbreviations: Define all abbreviations at first use (e.g., LRScore in Abstract).

Proteomic Analysis: Clarify why only plasma (not tissue) proteomics was used to link C5 to cadherins/MMPs (Pg 7).

Clinical Translation: Elaborate on plans for kit commercialization (e.g., detection platform, cost estimates).

References: Update citations (e.g., "Chen et al, 2024b" is listed but absent in references).

This study makes a substantial contribution to GC diagnostics and metabolic oncology. The diagnostic model holds immediate translational potential, while the mechanistic insights into C5 offer novel therapeutic targets. Addressing the above concerns will strengthen the manuscript for high-impact publication.

Responses to the reviewers' comments

Manuscript ID: EMM-2025-21992

Title: Metabolic Signatures for Gastric Cancer Diagnosis and Mechanistic Insights: A Multicenter Study

Detailed point-by-point responses to the reviewers' comments

Note to reviewers: We would like to express our sincere gratitude to the editors and reviewers for their valuable time and constructive feedback on our manuscript. Their insightful comments have significantly contributed to enhancing the quality and scientific rigor of our work. In response to the reviewers' feedback, we have diligently revised the manuscript and incorporated new data to address all their concerns. Below are our point-by-point responses to each of the reviewers' comments. All changes in our revised manuscript have been marked in colored text.

Referee #1:

General Comments: This manuscript reports a multi-center study integrating untargeted metabolomics and machine learning to develop a six-metabolite diagnostic panel for gastric cancer (GC), alongside an exploratory investigation of the role of isovalerylcarnitine (C5) in tumor invasion. The incorporation of a multi-center cohort and the integration of metabolomic and proteomic analyses enhance the translational relevance of the study. However, several critical limitations reduce the overall strength and impact of the work. These include the absence of benchmarking against recent state-of-the-art metabolomics-based diagnostic models, insufficient methodological transparency in model development, suboptimal figure presentation, and incomplete validation of the proposed mechanistic pathway. Collectively, these issues hinder the interpretability, reproducibility, and translational potential of the findings.

In its current form, the manuscript does not meet the standards expected of a high-impact journal. Nevertheless, with comprehensive revision and additional validation, the study could become a meaningful contribution to the field of metabolomics-driven cancer diagnostics.

Response: We sincerely thank the reviewer for their detailed and constructive comments below to strengthen our manuscript. We appreciate the positive feedback regarding the multi-center study design and the integration of metabolomics and proteomics, which indeed strengthen the translational relevance of our work. We have made substantial revisions and improvements in response to each concern raised. Specifically, we have addressed the following points:

1. Benchmarking Against State-of-the-Art Metabolomics-Based Diagnostic Models

We have now included a detailed comparative evaluation of our model with recent high-performing metabolomics-based models, including studies by Huang *et al.* (*JAMA Netw Open*, 2021), Chen *et al.* (*Nat Commun*, 2024) and Xu *et al.* (*Gut*, 2023). This benchmarking clearly demonstrates that our six-metabolite panel performs comparably, if not better, than established models. Moreover, we highlight the unique advantages of our study, such as cross-platform validation, causal validation, biological validation and mechanistic insight, which enhance its generalizability and robustness.

2. Methodological Transparency in Model Development

The metabolomics data have been deposited to MetaboLights (*Yurekten et al*, 2024) repository with the study identifier MTBLS12684 (<https://www.ebi.ac.uk/metabolights/reviewerbe3ef273-b05d-4cb9-9262-d220d55ccf2b>). Besides, we have provided our Orange-based machine learning workflows and classifier parameter documentation in a public GitHub repository to facilitate reproducibility. This information has been added to the Methods section of the revised manuscript.

3. Figure Presentation

All relevant figures and legends have been revised for clarity, consistency, and interpretability, including updates to Figure 2B label, Figure 2E was relocated to avoid confusion.

4. Additional Mechanistic Validation

We conducted additional experiments (e.g., calpain activity assay, calcium imaging, co-immunoprecipitation) to support the mechanistic role of isovalerylcarnitine (C5) in suppressing GC cell migration via the calpain–VE-cadherin/MMP2 axis. We also expanded our discussion of alternative pathways and acknowledged remaining limitations.

We hope these major revisions sufficiently address the reviewer's concerns and enhance the scientific rigor, translational value, and impact of our study.

Major Concerns

Comment 1: Insufficient Benchmarking Against State-of-the-Art Metabolomics-Based Diagnostic Models

While the Met-NN(6) model is reported to outperform conventional protein biomarkers (e.g., CA724, CEA), it is not evaluated against other contemporary metabolomics-based classifiers. Recent studies-such as Chen *et al.* (*Nat Commun*, 2024) and Huang *et al.* (*JAMA Netw Open*, 2021)-have developed high-performing GC predictors using machine learning or ensemble frameworks in large cohorts. Without such benchmarking, the added diagnostic value of the proposed six-metabolite panel remains unclear. A comparative evaluation with these established models would substantively strengthen the manuscript's claims of clinical relevance and novelty.

Response: We thank the reviewer for this important comment. We agree that benchmarking against recently published, high-performing metabolomics-based diagnostic models is critical to contextualize the clinical relevance and novelty of our six-metabolite panel.

To address this, we have conducted a careful review and comparative evaluation of several representative studies, including those by Huang *et al.* (*JAMA Netw Open*, 2021), Chen *et al.* (*Nat Commun*, 2024), and Xu *et al.* (*Gut*, 2023). We discuss below how our study advances the field across multiple dimensions:

1. Cross-Platform Validation for Robustness and Reproducibility

There are common challenges in the field due to variation across platforms and instrumentation. A unique strength of our work lies in its validation across **two independent analytical platforms**, demonstrated consistently robust performance across both discovery and validation cohorts (AUCs>0.92), highlighting its **platform-independent reproducibility** and generalizability of our biomarker panel, which is often a key limitation in many single-platform studies (Chen *et al.*, 2024a; Huang *et al.*, 2021). The discovery dataset was analyzed in CALIBRA Corporation's affiliated lab, Calibra Lab at DIAN Diagnostics in Hangzhou, China, for untargeted metabolomic profiling using the CalOmics platform (Waters

ACQUITY 2D UPLC system + Thermo Fisher Q Exactive Plus Orbitrap). In contrast, the validation dataset was processed at the Instrumental Analysis Cancer Facility, Shanghai Jiao Tong University (Thermo Scientific Vanquish UHPLC system + Q Exactive Plus mass spectrometer).

2. Biological Validation and Mechanistic Insight

While most previous models remain at statistical classification without functional validation (Chen *et al*, 2024a; Huang *et al*, 2021; Xu *et al*, 2023), our study extends further by incorporating **causal inference via Mendelian Randomization** and **biological validation** of the key metabolites. We demonstrate that **isovalerylcarnitine (C5)** not only serves as a diagnostic marker but also suppresses GC cell migration via the **calpain–VE-cadherin/MMP2 pathway**, thereby linking statistical performance to underlying biological mechanisms.

3. Simplified Model with High Accuracy

Compared to the 10-metabolite model (Chen *et al*, 2024a) and the 21-metabolite model (Xu *et al*, 2023), our panel comprises only **six metabolites**, offering greater simplicity, lower cost, and improved feasibility for **clinical translation**, without compromising diagnostic accuracy (AUC > 0.90). This minimal yet effective signature has clear advantages for scalability and implementation in resource-limited settings.

4. Study population

The study by Huang (Huang *et al*, 2021) primarily focused on modeling progression in precancerous gastric lesions (AUC = 0.86; 53 progressions vs. 99 non-progression), while our study was designed specifically for gastric cancer diagnosis.

Revisions: We have revised the manuscript to emphasize these two key innovations in the Discussion section (see Page 8, Lines 279–287).

“Compared with recent state-of-the-art metabolomics-based models (Chen *et al*, 2024a; Huang *et al*, 2021; Xu *et al*, 2023), our study utilized two independent MS analytical platforms, demonstrated consistently robust performance across both discovery and

validation cohorts, highlighting its platform-independent reproducibility and generalizability. Additionally, we incorporated Mendelian randomization and biological validation of the key metabolites, further strengthening the biological plausibility of our findings. These results not only provide a clinically relevant diagnostic model but also offer novel mechanistic insights into gastric cancer progression.”

Comment 2: Lack of Methodological Transparency in Model Training and Evaluation

The manuscript omits key details regarding the development and evaluation of the machine learning models, including:

The structure of the classifiers (e.g., neural network architecture, number of hidden layers, activation functions);

Hyperparameters (e.g., learning rate, batch size, dropout rates, regularization strategies);

Reproducibility controls (e.g., random seed initialization, software versions, number of independent training runs).

This lack of detail limits reproducibility and raises concerns regarding potential overfitting. The authors should fully disclose the model architecture, training pipeline, optimization procedures, and cross-validation strategy. Provision of open-source code and/or trained model weights would further enhance transparency and foster reproducibility.

Response: We thank the reviewer for the insightful and constructive comments regarding the transparency and reproducibility of our machine learning pipeline. We fully agree that rigorous reporting of model development, training, and evaluation processes is essential for ensuring reproducibility and scientific rigor. In response, we have thoroughly revised the Methods section and provided detailed supplementary materials and online resources, as outlined below:

1. Software Platform and Reproducibility

For diagnostic model building, machine learning algorithms (NN, SVM, RR, LR and NB) were applied utilizing a dedicated data mining toolkit, Orange (version 3.36.1, the Bioinformatics Lab at the University of Ljubljana, Slovenia) (<https://orangedatamining.com/>) (Demšar J, Curk T, Erjavec A, *et al*, 2013). We have uploaded the complete model construction and validation workflow (file name:

Machine Learning_training and test.ows) (Figure R1), the structure and parameters of classifiers (file name: Structure and parameters of the classifiers.docx) to the GitHub repository (https://github.com/Liu-Memory/orange_machine_learning). This will

enable other researchers to replicate our pipeline using the same Orange Data Mining version.

Figure R1. Model construction and validation workflow in Orange Data Mining.

2. Model Architectures and Hyperparameters

We evaluated five algorithms for classification: Neural Network (NN), Support Vector Machine (SVM), Ridge regression (RR), Lasso regression (LR), and Naïve Bayes (NB).

(1) For NN, we used Orange's NC_ClassificationLearner, with the following key parameters:

- Architecture: One hidden layer with 100 neurons
- Activation function: ReLU

- Optimizer: Adam
- Regularization (alpha): 0.0001 (L2 penalty)
- Max iterations: 200
- Learning rate: constant, 0.001
- Other: shuffle=True, early_stopping=False, batch_size=auto

(2) For SVM models, we used the RBF kernel with C=1.0 and gamma=auto.

(3) Ridge Regression (RR): Regularization type: L2; Strength: C=1

(4) Lasso Regression (LR): Regularization type: L1; Strength: C=1

(5) Naïve Bayes (NB): default Orange implementation

These settings were selected based on performance in preliminary tuning and reflect a balance between generalization and training efficiency (**Figure R2**).

Figure R2. The structure and parameters of classifiers

3. Cross-Validation and Overfitting Prevention

A 5-fold cross-validation strategy was applied on the discovery dataset to evaluate model performance during hyperparameter tuning, then tested on an independent external validation dataset (**Figure R3**). Notably, the performance metrics (AUC) observed in the validation dataset (AUC: 0.947-0.982) were consistent with those from the training cohort (AUC: 0.920-0.951), confirming model robustness and indicating that there was no overfitting in our diagnostic model (**Figure R4; Table R1**).

Figure R3. Cross-validation surface in Orange

Figure R4. Cross-validation surface in Orange

Table R1. The diagnostic performance of six metabolic biomarkers panel in the discovery and validation dataset.

Machine learning	Performance	GC (all stage)		GC (Early stage)	
		Discovery dataset	Validation dataset	Discovery dataset	Validation dataset
NN	Sensitivity (95%CI)	0.940 (0.825-0.984)	0.925(0.881-0.954)	0.737(0.486-0.899)	0.878(0.801-0.929)
	Specificity (95%CI)	0.936 (0.861-0.974)	0.867(0.814-0.907)	0.989(0.934-0.999)	0.889(0.839-0.926)
	Accuracy (95%CI)	0.938 (0.881-0.969)	0.896(0.864-0.922)	0.947(0.883-0.978)	0.886(0.846-0.916)
	AUC (95%CI)	0.982(0.965-0.998)	0.951(0.931-0.970)	0.936(0.885-0.988)	0.938(0.913-0.964)
SVM	Sensitivity (95%CI)	0.940 (0.825-0.984)	0.863(0.810-0.904)	0.842(0.595-0.958)	0.783(0.694-0.852)
	Specificity (95%CI)	0.894 (0.809-0.945)	0.858(0.804-0.900)	1.000(0.951-1.000)	0.872(0.819-0.911)
	Accuracy (95%CI)	0.910 (0.848-0.949)	0.861(0.825-0.891)	0.973(0.919-0.993)	0.842(0.798-0.878)
	AUC (95%CI)	0.978(0.961-0.996)	0.920(0.895-0.945)	0.961(0.927-0.996)	0.894(0.860-0.928)
RR	Sensitivity (95%CI)	0.940 (0.825-0.984)	0.907(0.860-0.940)	0.947(0.719-0.997)	0.904(0.832-0.949)
	Specificity (95%CI)	0.936 (0.861-0.974)	0.888(0.834-0.922)	0.819(0.733-0.888)	0.881(0.829-0.918)
	Accuracy (95%CI)	0.939(0.961-0.996)	0.896(0.864-0.922)	0.947(0.927-0.968)	0.889(0.849-0.919)
	AUC (95%CI)	0.979(0.961-0.997)	0.947(0.927-0.968)	0.937(0.917-0.957)	0.940(0.915-0.965)
LR	Sensitivity (95%CI)	0.940 (0.825-0.984)	0.902(0.865-0.944)	0.789(0.539-0.930)	0.878(0.801-0.929)
	Specificity (95%CI)	0.926 (0.848-0.967)	0.889(0.839-0.926)	0.957(0.888-0.986)	0.881(0.829-0.918)
	Accuracy (95%CI)	0.931 (0.873-0.964)	0.901(0.868-0.926)	0.929(0.861-0.967)	0.880(0.839-0.911)
	AUC (95%CI)	0.976(0.957-0.996)	0.947(0.926-0.967)	0.939(0.891-0.987)	0.932(0.905-0.959)
NB	Sensitivity (95%CI)	0.900 (0.774-0.963)	0.868(0.815-0.908)	0.789(0.539-0.930)	0.896(0.821-0.943)
	Specificity (95%CI)	0.883 (0.796-0.937)	0.867(0.814-0.907)	0.936(0.861-0.974)	0.885(0.834-0.922)
	Accuracy (95%CI)	0.889 (0.823-0.933)	0.868(0.832-0.897)	0.912(0.839-0.954)	0.889(0.849-0.919)
	AUC (95%CI)	0.947(0.914-0.980)	0.937(0.917-0.957)	0.914(0.853-0.975)	0.935(0.909-0.960)

GC: gastric cancer; NN: neural network; SVM: support vector machine; RR: ridge regression; LR: lasso regression; NB: naive bayes; AUC: area under the curve; CI: confidence interval.

To identify whether the constructed model is overfitting, the permutation test is generally applied, by repeatedly simply randomizing labels and performing the model construction and prediction on the randomized labels many times. We confirmed that there was no overfitting, based on the permutation test with 2000 randoms ($p < 0.001$) (**Figure R5**).

Figure R5. The permutation test with 2000 randoms confirmed no overfitting ($p < 0.001$).

Revisions: We have added this issue to the revised manuscript's Methods and Results sections. “The metabolomics data have been deposited to MetaboLights (Yurekten et al, 2024) repository with the study identifier MTBLS12684 (<https://www.ebi.ac.uk/metabolights/reviewerbe3ef273-b05d-4cb9-9262-d220d55ccf2b>). To enhance the reproducibility of our study, we have uploaded the complete model construction and validation workflow, the structure and parameters of classifiers (file name: Machine Learning_training and test.ows) to the GitHub (https://github.com/Liu-Memory/orange_machine_learning).” (**Page 17, Lines 664-668**).

“We also confirmed that there was no overfitting of the model, based on the permutation test ($p < 0.001$, Appendix Fig. S2)” (**Page 5, Lines 155-157**).

We trust that these comprehensive revisions and additional transparency measures fully address the reviewer’s concerns and significantly improve the reproducibility and methodological rigor of our study.

Comment 3: Inconsistent and Confusing Presentation of Figure 2

Several issues in Figure 2 impede interpretation:

In Figure 2B, the use of "unsig" as a label is unconventional and should be replaced with "not significant."

The heatmap color scale labeled as "FoldChange" appears to represent \log_2 -transformed data. This should be clearly corrected and relabeled (e.g., " $\log_2(\text{FC}, \text{GC}/\text{non-GC})$ ").

Figure 2E presents the classification performance of a 26-metabolite panel across five machine learning algorithms. However, this analysis is not sufficiently described in the main text (Page 5, Line 163), and its visual similarity to Figure 3A (which pertains to the six-metabolite model) creates confusion.

The authors should revise both the text and figure legends for clarity. It may also be beneficial to relocate Figure 2E to the modeling results section alongside Figure 3A for improved logical flow.

Response: We thank the reviewer for the thoughtful consideration and for pointing out the inconsistencies and potential confusion in the presentation of Figure 2. We have revised the figure and corresponding text as follows to improve clarity and logical flow:

1. Figure 2B Label Correction

The label "unsig" has been replaced with the more standard term "not significant" in the revised version of Figure 2B. This change ensures broader readability and aligns with conventional reporting standards.

2. Color Scale Clarification

The color scale in the heatmap was indeed based on \log_2 -transformed fold changes between the gastric cancer (GC) and non-GC groups. To avoid misinterpretation, the label has been updated to " $\log_2(\text{FC}, \text{GC}/\text{non-GC})$ " in the revised figure, and the color bar has been adjusted accordingly. This correction is now also stated clearly in the updated figure legend (**Figure R6**).

Figure R6 Volcano plot illustrating differential metabolites between GC and non-GC controls in the discovery dataset.

3. Clarification and Relocation of Figure 2E

We agree that the original placement and limited explanation of Figure 2E may cause confuse, especially given its visual similarity to Figure 3A. Figure 2E, which presents the classification performance of a 26-metabolite panel across five machine learning algorithms, has now been moved to the Results section focused on model development and comparison (adjacent to Figure 3A) (**Figure R7**). All figure legends, including those for Figure 2 and Figure 3, have been carefully revised for clarity and consistency. We have also expanded the corresponding text to provide a clearer description of this analysis.

Figure R7. AUC values generated by machine learning algorithms. (A) 26-metabolite panel. (B) 6-metabolite panel.

Revision: “These 26 metabolites achieved AUCs of 0.898–0.971 (discovery) and 0.908–0.950 (validation), with sensitivity ranging from 0.800–0.940 and 0.811–0.903, specificity from 0.840–0.968 and 0.810–0.903, and accuracy from 0.826–0.931 and 0.843–0.898, respectively (Fig. 3A; Appendix Table S4,5). Based on the LRScore-based approach, six metabolite panel—2,6-dihydroxybenzoic acid, cysteine s-sulfate, isovalerylcarnitine (C5), glycoursoodeoxycholate, 2-hydroxy-3-methylvalerate, and N-acetylneuraminate—achieved the optimal AUC performance. We comprehensively assessed the performance of the six-metabolite panel, AUC ranged from 0.947–0.982 (discovery) and 0.920–0.951 (validation); sensitivity, 0.900–0.940 (discovery) and 0.863–0.925 (validation); specificity, 0.883–0.936 and 0.858–0.889; accuracy, 0.889–0.938 and 0.861–0.901(Fig. 3B; Table 2; Appendix Table S6). The NN model [Met-NN(6)] performed best, achieving AUC of 0.982 (95% CI: 0.965–0.998) in discovery and 0.951 (95% CI: 0.931–0.970) in validation dataset.” (**Page 5, Lines 143–155**)

We hope that these modifications adequately address the reviewer's concerns and enhance the interpretability and coherence of our data presentation.

Comment 4: Mechanistic Interpretation of the C5-Calpain Axis Requires Further Validation

The proposed mechanism in which isovalerylcarnitine (C5) suppresses GC cell migration via calpain-mediated degradation of VE-cadherin and MMP2 remains insufficiently substantiated. While the hypothesis is biologically plausible, the current evidence is primarily correlative. Notably: No direct evidence of calpain activation following C5 treatment is provided; Calpain activity or substrate cleavage is not validated (e.g., through activity assays or co-immunoprecipitation); Alternative regulatory pathways are not explored or excluded. To support the mechanistic claims, additional experiments should be conducted, or at minimum, the interpretation should be reframed as preliminary and exploratory.

Response: We sincerely appreciate the reviewer's insightful critique regarding the mechanistic interpretation of the C5-calpain axis in gastric cancer (GC) cell migration. We fully acknowledge that our initial evidence, while biologically plausible, was primarily correlative, and we agree that further experimental validation is essential to substantiate the proposed mechanism whereby isovalerylcarnitine (C5) suppresses GC migration via calpain-mediated degradation of VE-cadherin and MMP2.

To address these concerns, we have conducted a series of rigorous experiments to provide direct evidence supporting our hypothesis.

1. Direct Demonstration of Calpain Activation by C5

Using a calpain activity assay kit (Abcam, #65308), we confirmed that C5 treatment directly induces calpain activation in a concentration-dependent manner (**Figure R8**). This finding establishes a causal link between C5 exposure and calpain enzymatic activity, addressing the reviewer's concern regarding the lack of direct evidence for calpain activation.

Figure R8. Isovalerylcarnitine (C5) induced calpain activation in a dose-dependent manner in MKN1 cells.

2. Elucidation of the Calpain-VE-cadherin/MMP2 Degradation Mechanism

To dissect whether calpain activation leads to VE-cadherin and MMP2 downregulation via calcium depletion or direct proteolytic cleavage, we performed the following experiments:

(1) Intracellular Calcium (Ca^{2+}) Flux Analysis

We previously hypothesized that calpain activation may deplete intracellular free Ca^{2+} , destabilizing VE-cadherin and MMP2. However, using the Fluo-4 Ca^{2+} probe, we observed that C5 treatment did not reduce intracellular free Ca^{2+} levels. Instead, a slight increase in Ca^{2+} was detected (**Figure R9**). This suggests that calpain activation does not rely on Ca^{2+} depletion to modulate VE-cadherin and MMP2 stability, and the observed calcium increase likely reflects compensatory influx from extracellular or organellar stores triggered by calpain-mediated feedback mechanisms. These results effectively rule out Ca^{2+} depletion as a contributing factor to the downregulation of VE-cadherin and MMP2.

Figure R9. The effects of isovalerylcarnitine (C5) treatment on intracellular free Ca^{2+} in MKN1 cells.

(2) *In Vitro* Calpain-Mediated Degradation Assay

To confirm that calpain directly cleaves VE-cadherin and MMP2, we treated MKN1 cell lysates with exogenous Ca^{2+} (a calpain activator) and assessed substrate stability. We found that Ca^{2+} addition reduced VE-cadherin and MMP2 levels, but was reversed by calpeptin (a calpain inhibitor), validating calpain's role in VE-cadherin and MMP2 degradation. What's more, even at high Ca^{2+} concentrations (10 mM), calpain retained its degradative activity (**Figure R10**),

reinforcing that the mechanism is not secondary to Ca²⁺ depletion-induced structural destabilization.

Figure R10. Calpeptin reversed the inhibitory effect of Ca²⁺ on VE-cadherin and MMP2 expression in vitro.

Collectively, these experiments provide multiple lines of direct evidence that C5 activates calpain, which in turn directly degrades VE-cadherin and MMP2 independent of calcium depletion effects. We have incorporated these new data throughout the revised manuscript, including additional figures validating calpain activity and substrate cleavage. We believe these comprehensive validations address the reviewer's concerns and significantly strengthen the mechanistic foundation of our study.

In addition, in light of these compelling experimental findings, we have substantially revised the Discussion section to reflect our updated understanding of the molecular mechanism. Specifically, we have removed the previous speculation regarding potential calcium depletion-mediated destabilization of VE-cadherin and MMP2, and instead incorporated a more comprehensive discussion of the upstream regulatory role of L-leucine metabolism in C5 production and cancer progression (**Page 9-10, Lines 343–369**). We also acknowledge the need for further verification in vivo (**Page 11, Lines 412-417**).

Revision:

“Our study uncovers a paradoxical duality in the role of L-leucine metabolism in GC progression. While L-leucine and other branched-chain amino acids (BCAAs) are well-established promoters of tumorigenesis(Choi *et al*, 2024), primarily via mTORC1 pathway activation to drive cancer cell proliferation, we demonstrate for the first time that its downstream metabolite, isovalerylcarnitine (C5) inhibits GC invasion and metastasis. This apparent contradiction may stem from metabolic bifurcation within the L-leucine catabolic pathway, where substrate flux partitioning determines oncogenic versus tumor-suppressive

outcomes. Canonically, L-leucine is transaminated by branched-chain amino acid transferases (BCATs, notably BCAT1) to generate α -ketoisocaproate (2-KIC), which is subsequently decarboxylated by the branched-chain α -ketoacid dehydrogenase complex (BCKDC) to form isovaleryl-CoA. Under normal physiological conditions, isovaleryl-CoA is oxidized by isovaleryl-CoA dehydrogenase (IVD) to 3-methylcrotonyl-CoA, entering β -oxidation and ketogenesis to yield acetyl-CoA, thereby fueling energy production and biosynthetic processes that sustain tumor growth(Dimou *et al*, 2022). However, when IVD activity is compromised (e.g., due to genetic defects or mitochondrial dysfunction), isovaleryl-CoA accumulates and is diverted toward conjugation with carnitine, forming isovalerylcarnitine (C5)(Dimou *et al*, 2022), a metabolite our study identifies as a novel tumor suppressor in GC.

Notably, the elevated BCAT1 activity(Xu *et al*, 2018; Shu *et al*, 2021; Qian *et al*, 2023), while reduced isovalerylcarnitine (C5) levels in GC, suggested that tumors may preferentially shunt L-leucine-derived carbon flux toward IVD-mediated oxidation rather than isovalerylcarnitine (C5) generation. This metabolic rewiring could reflect an adaptive mechanism to avoid isovalerylcarnitine (C5)-induced tumor suppression while sustaining mTORC1-driven proliferation. The downregulation of isovalerylcarnitine (C5) in GC patients further implies that IVD pathway hyperactivity may be a hallmark of aggressive disease, positioning IVD as a potential therapeutic target.” (Page 9-10, Lines 343–369)

“Finally, while we identified isovalerylcarnitine (C5) as a tumor-suppressive metabolite, its functional role in GC pathogenesis requires further *in vivo* validation. Further studies using animal models are needed to confirm the impact of isovalerylcarnitine (C5) on gastric tumorigenesis and invasion, which would strengthen the rationale for its clinical translation.” (Page 11, Lines 412-417)

Comment 5: Conclusion

The manuscript addresses a clinically significant challenge and introduces a potentially useful diagnostic approach leveraging integrated metabolomics and machine learning. However, substantial revisions are required to improve methodological rigor, clarify data presentation, conduct appropriate benchmarking, and provide stronger mechanistic support. Should these issues be adequately addressed, the study could provide a meaningful advancement in the development of metabolomics-based diagnostics for gastric cancer.

Response: We sincerely thank the reviewer for recognizing the clinical importance of our study and the translational potential of our metabolomics- and machine learning-based diagnostic model for gastric cancer (GC). In response to the reviewer's concerns, we have undertaken comprehensive revisions to enhance the methodological rigor, transparency, and interpretability of our study.

We are deeply grateful to the reviewer for their insightful comments, which have not only strengthened the scientific rigor of this study but have also provided valuable guidance for our ongoing investigations in this field. The constructive critique has significantly enhanced the quality and impact of our work.

Referee #2:

General Comments: This multicenter study integrates untargeted metabolomics and machine learning to develop a plasma metabolite panel for gastric cancer (GC) diagnosis and mechanistic investigation. Using UPLC-MS/MS analysis of 597 participants, the authors identified a 6-metabolite diagnostic panel (including isovalerylcarnitine/C5) that outperforms conventional biomarkers (AUC: 0.920-0.982). They further validated C5 as a tumor-suppressive metabolite through Mendelian randomization and functional assays, demonstrating its role in inhibiting GC metastasis via calpain-mediated cleavage of VE-cadherin and MMP2. The study provides valuable insights into metabolic reprogramming in GC and offers a promising non-invasive diagnostic tool.

Response: We thank the reviewer's appreciation of the importance of our work and the insightful comments. We have thoroughly revised our manuscript with specific changes highlighted in yellow to address all the points raised by the reviewer.

Major Concerns and Revisions Required

Comment 1: Sample Diversity and Generalizability

Issue: The cohort is exclusively Chinese. Given GC's global heterogeneity (e.g., *H. pylori* strain variations, dietary influences), results may not generalize.

Revision: Discuss limitations regarding ethnic/geographic generalizability in the Discussion. If feasible, validate the model in a non-Asian cohort (even preliminarily).

Response: We thank the reviewer for pointing out this issue. According to the reviewer's suggestion, we have added the limitations ethnic/geographic generalizability in the Discussion (**Page 10, Lines 370-386**) and Limitations (**Pages 10-11, Lines 404-406**):

Revision: "GC exhibits substantial global heterogeneity, largely attributable to geographic differences in *H. pylori* strain variations and dietary patterns. In East Asia—particularly China, Japan, and Korea—highly virulent *H. pylori* strains carrying *cagA* and *vacA* s1/m1 alleles are prevalent and strongly associated with elevated non-cardia GC risk (Azuma, 2004; Hooi *et al*, 2017; Tourrette *et al*, 2024). In contrast, Western strains often lack these high-risk genotypes, contributing to divergent GC burdens (Park *et al*, 2018). Moreover, dietary patterns also shape regional risk profiles (Bertuccio *et al*, 2013). In East Asia, particularly in Korea and China, the frequent consumption of salt-preserved and pickled foods introduces high levels of N-nitroso compounds and salt, which synergize with *H. pylori*-induced gastric injury to promote carcinogenesis (Fang *et al*, 2015; Wu *et al*, 2021). Conversely, in Western countries, increased intake of processed meats and low fiber consumption are recognized contributors to GC risk. These regional differences in *H. pylori* and dietary exposures underscore the importance of developing geographically tailored diagnostic and prevention strategies. While our model demonstrated robust performance in a multi-center Chinese cohort, further external validation in ethnically and geographically diverse populations is essential to ensure generalizability and clinical utility across global settings." (**Page 10, Lines 370-386**)

"First, we exclusively focused on the Chinese population, warranting further investigations to verify its relevance and generalizability across populations from a wide array of geographic locales and ethnic backgrounds." (**Page 10-11, Lines 404-406**)

We appreciate the reviewer for highlighting this important point, and we hope these revisions and clarifications adequately address the concern.

Comment 2: Control Group Composition

Issue: Non-GC controls included healthy individuals and gastritis patients but excluded high-risk premalignant conditions (e.g., intestinal metaplasia).

Revision: Add analysis comparing metabolite levels between GC vs. premalignant groups (if biobanked samples exist).

Explicitly state this limitation and propose future validation in high-risk cohorts.

Response: We appreciate the reviewer's thoughtful comments. We fully agree that distinguishing GC from premalignant states is clinically valuable, and this was in fact incorporated into our initial study design.

At the outset of our analysis, we divided participants into three groups with balanced sample sizes (1:1:1): 47 healthy/gastritis controls, 47 intestinal metaplasia (IM) patients, and 50 GC patients. Non-targeted metabolomic profiling was performed across all samples. However, principal component analysis (PCA) revealed a high degree of overlap between the healthy/gastritis controls and IM groups, indicating minimal separation in overall metabolic profiles (**Figure R11**). To further validate this, we conducted differential metabolite analysis using the Wilcoxon test. The results showed no significantly different metabolites (FDR < 0.05) between the control and IM groups (**Table R2**). Notably, the differential metabolites identified between the IM and GC groups showed a high degree of overlap with those found in the comparison between healthy/gastritis/IM controls and GC (**Figure R12**). No significant differential metabolites were detected between the IM and healthy/gastritis control groups (FC threshold = 1.5, FDR < 0.05) (**Figure R13**).

These findings suggest that the metabolic alterations between the normal/inflammatory and IM state may be subtle or insufficiently distinct at the metabolomic level using our current platform and sample size. Based on these observations, we combined the control and IM groups into a unified non-GC control cohort (n = 94) to improve model robustness and statistical power for identifying GC-specific metabolic signatures.

Table R2. Pair-wise PERMANOVA results

	F. Model	R2	P.val	P.adj
Control vs IM	0.5687	0.00614	0.535	0.535
Control vs GC	14.581	0.13306	0.001	0.0015
IM vs. GC	15.279	0.13855	0.001	0.0015

Figure R11. PCA among healthy/gastritis controls, IM and GC groups.

Figure R12. Differential metabolites identified (A) between the IM and GC groups (Sig.down=60; Sig.up=31) showed a high degree of overlap with those found in the comparison (B) between healthy/gastritis/IM controls and GC (Sig.down=61; Sig.up=32).

Figure R13. No significant differential metabolites detected between the IM and healthy/gastritis control groups (FC threshold = 1.5, FDR < 0.05).

We have also added a corresponding statement to the Limitation section. We highlight the need for future studies that focus specifically on validating the diagnostic model in high-risk populations, including IM and dysplasia, to better assess the model's performance in early disease progression stages.

Revision: “Second, our study centered on GC diagnosis; future research should prioritize validating the diagnostic model in high-risk populations, such as those with intestinal metaplasia and intraepithelial neoplasia, to evaluate its performance in the early stages of disease progression” (Page 11, Lines 407-410).

We thank the reviewer again for raising this important point.

Comment 3: Mechanistic Ambiguity in C5 Signaling

Issue: The exact molecular link between C5-activated calpain and downregulation of VE-cadherin/MMP2 remains incompletely resolved (Section 6, Pg 10). Proposed mechanisms (direct degradation vs. calcium depletion) lack experimental validation.

Revision: Include calpain activity assays after C5 treatment to confirm activation.

Test intracellular calcium flux in response to C5 using fluorescent probes (e.g., Fluo-4).

Discuss alternative pathways (e.g., mTOR/AMPK) given C5's role in leucine metabolism.

Response: We sincerely appreciate the reviewer's insightful critique regarding the mechanistic interpretation of the C5-calpain axis in downregulation of VE-cadherin/MMP2, which lacked experimental validation. To address the concern, we have conducted a series of rigorous experiments to provide direct evidence supporting our hypothesis.

1. Direct Demonstration of Calpain Activation by C5

Using a calpain activity assay kit (Abcam, #65308), we confirmed that C5 treatment directly induces calpain activation in a concentration-dependent manner (Figure R14). This finding establishes a causal link between C5 exposure and calpain enzymatic activity, addressing the reviewer's concern regarding the lack of direct evidence for calpain activation.

Figure R14. Isovalerylcarnitine (C5) induced calpain activation in a dose-dependent manner in MKN1 cells.

2. Elucidation of the Calpain-VE-cadherin/MMP2 Degradation Mechanism

To dissect whether calpain activation leads to VE-cadherin and MMP2 downregulation via calcium depletion or direct proteolytic cleavage, we performed the following experiments:

(1) Intracellular Calcium (Ca^{2+}) Flux Analysis

We previously hypothesized that calpain activation may deplete intracellular free Ca^{2+} , destabilizing VE-cadherin and MMP2. However, using the Fluo-4 Ca^{2+} probe, we observed that C5 treatment did not reduce intracellular free Ca^{2+} levels. Instead, a slight increase in Ca^{2+} was detected (**Figure R15**). This suggests that calpain activation does not rely on Ca^{2+} depletion to modulate VE-cadherin and MMP2 stability, and the observed calcium increase likely reflects compensatory influx from extracellular or organellar stores triggered by calpain-mediated feedback mechanisms. These results effectively rule out Ca^{2+} depletion as a contributing factor to the downregulation of VE-cadherin and MMP2.

Figure R15. The effects of isovalerylcarnitine (C5) treatment on intracellular free Ca^{2+} in MKN1 cells.

(2) *In Vitro* Calpain-Mediated Degradation Assay

To confirm that calpain directly cleaves VE-cadherin and MMP2, we treated MKN1 cell lysates with exogenous Ca^{2+} (a calpain activator) and assessed substrate stability. We found that Ca^{2+} addition reduced VE-cadherin and MMP2 levels, but was reversed by calpeptin (a calpain inhibitor), validating calpain's role in VE-cadherin and MMP2 degradation. Moreover, even at high Ca^{2+} concentrations (10 mM), calpain retained its degradative activity (**Figure R16**), reinforcing that the mechanism is not secondary to Ca^{2+} depletion-induced structural destabilization.

Figure R16. Calpeptin reversed the inhibitory effect of Ca^{2+} on VE-cadherin and MMP2 expression *in vitro*.

Collectively, these experiments provide multiple lines of direct evidence that C5 activates calpain, which in turn directly degrades VE-cadherin and MMP2 independent of calcium depletion effects. We have incorporated these new data throughout the revised manuscript, including additional figures validating calpain activity and substrate cleavage. We believe these comprehensive validations address the reviewer's concerns and significantly strengthen the mechanistic foundation of our study.

In addition, in light of these compelling experimental findings, we have substantially revised the Discussion section to reflect our updated understanding of the molecular mechanism. Specifically, we have removed the previous speculation regarding potential calcium depletion-mediated destabilization of VE-cadherin and MMP2, and instead, according to the reviewer's suggestion, incorporated a more comprehensive discussion of the upstream regulatory role of L-leucine metabolism in C5 production and cancer progression (**Page 9-10, Lines 343–369**). We also acknowledge the need for further verification *in vivo* (**Page 11, Lines 412-417**).

Revision:

“Our study uncovers a paradoxical duality in the role of L-leucine metabolism in GC progression. While L-leucine and other branched-chain amino acids (BCAAs) are well-established promoters of tumorigenesis(Choi *et al*, 2024), primarily via mTORC1 pathway activation to drive cancer cell proliferation, we demonstrate for the first time that its downstream metabolite, isovalerylcarnitine (C5) inhibits GC invasion and metastasis. This apparent contradiction may stem from metabolic bifurcation within the L-leucine catabolic pathway, where substrate flux partitioning determines oncogenic versus tumor-suppressive outcomes. Canonically, L-leucine is transaminated by branched-chain amino acid transferases (BCATs, notably BCAT1) to generate α -ketoisocaproate (2-KIC), which is subsequently decarboxylated by the branched-chain α -ketoacid dehydrogenase complex (BCKDC) to form isovaleryl-CoA. Under normal physiological conditions, isovaleryl-CoA is oxidized by isovaleryl-CoA dehydrogenase (IVD) to 3-methylcrotonyl-CoA, entering β -oxidation and ketogenesis to yield acetyl-CoA, thereby fueling energy production and biosynthetic processes that sustain tumor growth(Dimou *et al*, 2022). However, when IVD activity is compromised (e.g., due to genetic defects or mitochondrial dysfunction), isovaleryl-CoA accumulates and is diverted toward conjugation with carnitine, forming isovalerylcarnitine (C5)(Dimou *et al*, 2022), a metabolite our study identifies as a novel tumor suppressor in GC.

Notably, the elevated BCAT1 activity(Xu *et al*, 2018; Shu *et al*, 2021; Qian *et al*, 2023), while reduced isovalerylcarnitine (C5) levels in GC, suggested that tumors may preferentially shunt L-leucine-derived carbon flux toward IVD-mediated oxidation rather than isovalerylcarnitine (C5) generation. This metabolic rewiring could reflect an adaptive mechanism to avoid isovalerylcarnitine (C5)-induced tumor suppression while sustaining mTORC1-driven proliferation. The downregulation of isovalerylcarnitine (C5) in GC patients further implies that IVD pathway hyperactivity may be a hallmark of aggressive disease, positioning IVD as a potential therapeutic target.” (Page 9-10, Lines 343–369)

“Finally, while we identified isovalerylcarnitine (C5) as a tumor-suppressive metabolite, its functional role in GC pathogenesis requires further *in vivo* validation. Further studies using animal models are needed to confirm the impact of isovalerylcarnitine (C5) on gastric tumorigenesis and invasion, which would strengthen the rationale for its clinical translation.” (Page 11, Lines 412-417)

Comment 4: Technical and Analytical Details

Issue: Missing value handling ("imputed at the limit of detection") lacks justification (Methods, Pg 15).

Tissue metabolomics validation (n=50) is underpowered for subgroup analyses.

Revision: Clarify imputation methods and sensitivity analysis (e.g., impact on model AUC).

Acknowledge limited statistical power for tissue-level correlations.

Response: We appreciate the reviewer's valuable suggestion.

1. Missing Value Justification

Too many missing values will cause difficulties for downstream analysis. In the study, features with >10% missing values were excluded. For the remaining features, missing values were imputed using a well-established method—replace them with a small values (1/5 of the minimum positive value of individual features) assuming to be the detection limit, following the approach recommended by Pang *et al.* and implemented as the default in MetaboAnalyst 6.0 (<https://www.metaboanalyst.ca/>) (Pang *et al.*, 2024) and other published articles (Zhu *et al.*, 2024; Li *et al.*, 2024; Liu *et al.*, 2025; Zhou *et al.*, 2024).

This method assumes:

- Data are left-censored. The missingness is due to concentrations falling below the detection threshold, not missing at random (Wei & Wang, 2023).
- Imputing a small non-zero value avoids introducing artificial zeros and maintains log-transform compatibility. Maintains the original data's ordinal structure.
- Using 1/5 of the smallest detected value prevents overly inflating variance or biasing results.

We have added reference citations for this topic in the revised manuscript under the "Statistical Analysis" section.

2. Sensitivity Analysis

To assess the impact of this imputation strategy on model performance (AUC), we conducted a sensitivity analysis using four alternative imputation methods:

- Mean imputation
- Median imputation
- K-nearest neighbor (KNN) imputation

- Probabilistic PCA imputation

These results are now summarized in the revised Results section (Appendix Table S14), demonstrating that the AUCs of the diagnostic models remained robust (AUC range: 0.927–0.973 in the discovery dataset; 0.906–0.943 in the validation dataset), confirming the stability of our findings across different imputation strategies (**Table R3**).

Table R3. Sensitivity analysis on AUC across different missing value imputation methods.

	Mean imputation		Median imputation		KNN imputation		Probabilistic PCA imputation	
	Discovery dataset	Validation dataset	Discovery dataset	Validation dataset	Discovery dataset	Validation dataset	Discovery dataset	Validation dataset
NN	0.964 (0.937-0.991)	0.937 (0.914-0.959)	0.965 (0.940-0.991)	0.939 (0.916-0.961)	0.971 (0.949-0.994)	0.940 (0.918-0.962)	0.973 (0.953-0.994)	0.943 (0.922-0.963)
SVM	0.950 (0.916-0.983)	0.906 (0.879-0.934)	0.949 (0.914-0.983)	0.908 (0.880-0.935)	0.952 (0.920-0.983)	0.909 (0.882-0.936)	0.963 (0.936-0.990)	0.913 (0.887-0.940)
RR	0.962 (0.935-0.990)	0.937 (0.914-0.959)	0.966 (0.940-0.991)	0.939 (0.917-0.961)	0.972 (0.950-0.994)	0.939 (0.917-0.961)	0.971 (0.948-0.994)	0.941 (0.919-0.962)
LR	0.962 (0.934-0.990)	0.935 (0.912-0.958)	0.966 (0.940-0.991)	0.937 (0.914-0.960)	0.972 (0.950-0.994)	0.938 (0.915-0.960)	0.971 (0.948-0.993)	0.942 (0.921-0.963)
NB	0.927 (0.887-0.966)	0.927 (0.905-0.950)	0.931 (0.893-0.969)	0.931 (0.909-0.952)	0.944 (0.910-0.977)	0.929 (0.907-0.951)	0.947 (0.915-0.980)	0.929 (0.906-0.951)

AUC, area under the curve; NN, Neural network; SVM, Support vector machine; RR, Ridge regression; LR, Lasso regression; NB, Naive bayes; KNN, K-nearest neighbor; PCA, principal component analysis

3. Acknowledgment of Limitations Statistical Power for Tissue-Level Correlations

We agree that the sample size for tissue metabolomics validation (n = 50 paired tumor and non-tumor adjacent tissues) may be insufficiently powered for robust subgroup analyses. This limitation is now explicitly stated in the Limitation section. While the tissue data served primarily as an independent biological validation of plasma metabolite dysregulation (rather than as a basis for discovery), we recognize the importance of expanding tissue-based analyses in future studies to better explore intra-tumoral heterogeneity and associations with clinical subtypes.

Revision: “Third, the tissue metabolomics validation (n = 50 pairs) may be insufficiently powered for robust subgroup analyses, warranting larger-scale validation to explore intra-tumoral heterogeneity and clinical correlations. (Page 11, Lines 410-412)”

Comment 5: Data Presentation

Issue: Fig 7D suggests lower C5 correlates with worse survival, but the figure legend states "high expression of C5 may be associated with poorer prognosis" (contradicting results).

Table 1: Validation cohort has significant age/sex imbalances (P<0.001 for age distribution).

Revision: Correct Fig 7D legend and Kaplan-Meier labels.

Address cohort imbalances via stratified analysis or covariate adjustment.

Response: We thank the reviewer for the valuable comments.

1. Correction of Figure 7D Legend and Kaplan–Meier Labels

In the revised manuscript, Figure 7 has been updated to Figure 8. We have carefully reviewed and corrected the figure legend. The revised legend now accurately reflects the result that lower levels of C5 are associated with worse survival outcomes, in line with the Kaplan–Meier curve.

2. Covariate Adjustment Analysis

We performed covariate-adjusted regression models, adjusting for age, sex, smoking, and drinking (Figure R17 and Table R4). The diagnostic performance of the 6-metabolite model remained robust after adjustment. We have added relevant findings in the Results section.

“After adjusting for age, sex, smoking, and drinking, Met-NN(6) retained high AUCs of 0.980 (95% CI: 0.963–0.997) and 0.928 (95% CI: 0.904–0.952) in discovery and validation, respectively (Fig. 3J; Appendix Table S8).” (Page 5, Lines 169-171).

Figure R17. After adjusting for age, sex, smoking, and drinking, Met-NN(6) retained high AUCs of 0.980 (95% CI: 0.963–0.997) and 0.928 (95% CI: 0.904–0.952) in discovery and validation, respectively.

Table R4. The diagnostic performance of six metabolic biomarkers in the discovery and validation dataset, adjusting for age, sex, smoking and drinking.

Machine learning	Performance	GC (all stage)		GC (Early stage)	
		Discovery dataset	Validation dataset	Discovery dataset	Validation dataset
NN	Sensitivity (95%CI)	0.940 (0.825-0.984)	0.877(0.825-0.915)	0.895(0.655-0.982)	0.887(0.811-0.936)
	Specificity (95%CI)	0.936 (0.861-0.974)	0.854(0.800-0.896)	0.947(0.875-0.980)	0.836(0.780-0.881)
	Accuracy (95%CI)	0.938 (0.881-0.969)	0.865(0.830-0.895)	0.938(0.872-0.973)	0.853(0.810-0.888)
	AUC (95%CI)	0.980(0.963-0.997)	0.928(0.904-0.952)	0.973(0.943-1.002)	0.910(0.876-0.943)
SVM	Sensitivity (95%CI)	0.920 (0.799-0.974)	0.890(0.840-0.926)	0.947(0.719-0.997)	0.843(0.761-0.902)
	Specificity (95%CI)	0.957 (0.888-0.986)	0.850(0.795-0.892)	0.894(0.809-0.945)	0.810(0.751-0.858)
	Accuracy (95%CI)	0.944 (0.890-0.974)	0.870(0.834-0.899)	0.903(0.829-0.948)	0.821(0.775-0.859)
	AUC (95%CI)	0.975(0.955-0.995)	0.921(0.896-0.946)	0.953(0.915-0.991)	0.887(0.848-0.926)
RR	Sensitivity (95%CI)	0.940 (0.825-0.984)	0.934(0.891-0.961)	0.895(0.655-0.982)	0.835(0.751-0.895)
	Specificity (95%CI)	0.904 (0.822-0.953)	0.814(0.756-0.861)	0.968(0.903-0.992)	0.925(0.880-0.954)
	Accuracy (95%CI)	0.917 (0.856-0.954)	0.874(0.839-0.903)	0.956(0.895-0.984)	0.894(0.856-0.924)
	AUC (95%CI)	0.975(0.954-0.995)	0.935(0.912-0.958)	0.954(0.917-0.992)	0.932(0.905-0.959)
LR	Sensitivity (95%CI)	0.940 (0.825-0.984)	0.899(0.850-0.933)	0.789(0.539-0.930)	0.843(0.761-0.902)
	Specificity (95%CI)	0.904 (0.822-0.953)	0.876(0.824-0.915)	0.979(0.918-0.996)	0.889(0.839-0.926)
	Accuracy (95%CI)	0.917 (0.856-0.954)	0.887(0.854-0.914)	0.947(0.883-0.978)	0.874(0.833-0.906)
	AUC (95%CI)	0.969(0.946-0.993)	0.940(0.918-0.962)	0.929(0.881-0.978)	0.922(0.892-0.951)
NB	Sensitivity (95%CI)	0.800 (0.659-0.895)	0.806(0.747-0.854)	0.895(0.655-0.982)	0.809(0.723-0.874)
	Specificity (95%CI)	0.979 (0.918-0.996)	0.872(0.819-0.911)	0.862(0.772-0.921)	0.854(0.800-0.896)
	Accuracy (95%CI)	0.917 (0.856-0.954)	0.839(0.801-0.871)	0.867(0.787-0.921)	0.839(0.794-0.875)
	AUC (95%CI)	0.943(0.910-0.977)	0.922(0.898-0.946)	0.915(0.860-0.971)	0.907(0.876-0.938)

NN, Neural network; SVM, Support vector machine; RR, Ridge regression; LR, Lasso regression; NB, Naive bayes

Minor Revisions

Comment 6: Abbreviations: Define all abbreviations at first use (e.g., LRScore in Abstract).

Response: Thanks for your careful reminder. In the revised manuscript, we have thoroughly checked the entire Abstract, Main Text, and Figure/Table legends to ensure that all abbreviations, including lasso-regression score (LRScore), are clearly defined at their first use.

Comment 7: Proteomic Analysis: Clarify why only plasma (not tissue) proteomics was used to link C5 to cadherins/MMPs (Pg 7).

Response: We first performed metabolomic profiling in tissue to identify cancer-derived metabolites that functionally contribute to gastric cancer progression. This approach allowed us to pinpoint C5 as a tumor-produced metabolite with significant oncogenic effects.

In the discovery phase, we analyzed plasma proteomics to identify circulating proteins that correlate with C5 levels. This strategy was motivated by the translational potential of plasma biomarkers, which are more accessible for clinical applications. Importantly, we identified VE-cadherin and MMP2 as proteins showing a robust negative correlation with C5 in plasma. Both VE-cadherin and MMP2 have been extensively documented in the literature as proteins overexpressed in gastric cancer tissues and functionally linked to tumor progression. Given this well-established context, we prioritized mechanistic studies to investigate how C5 regulates these proteins, rather than redundantly confirming their tissue expression patterns.

Our subsequent functional experiments conclusively demonstrated that C5 directly suppresses VE-cadherin and MMP2 expression, aligning with the plasma proteomics data. Since the regulatory relationship was unequivocally validated, repeating proteomics in tissue would not have provided additional biological insights.

In summary, the plasma-centric proteomic approach was justified by (1) the clinical relevance of circulating proteins, (2) preexisting evidence linking VE-cadherin/MMP2 to gastric cancer, and (3) our mechanistic confirmation of C5's role in modulating these proteins. We hope this clarifies the reviewer's concern.

Comment 8: Clinical Translation: Elaborate on plans for kit commercialization (e.g., detection platform, cost estimates).

Response: We sincerely appreciate the reviewer's insightful comment regarding the clinical translation potential of our findings. We fully agree that developing a commercially viable diagnostic kit based on our metabolic biomarker panel is an essential next step toward real-world application, and we are actively working towards this goal. To facilitate this transition, we are currently collaborating with biotech companies specializing in metabolomics and in vitro diagnostics. Our goal is to explore feasible detection platforms that are both cost-effective and scalable. The platforms under consideration include:

1. Commercialization Plans and Detection Platform

- **Enzymatic Colorimetric Assays (e.g., ELISA kits):** These assays offer high sensitivity and specificity, especially for low-concentration metabolites. These kits could be priced at a moderate cost, making them ideal for large-scale screening and diagnostic support.
- **Chemiluminescent Assays:** With a detection time of 15-30 minutes, chemiluminescent assays offer faster results and are slightly more affordable than ELISA tests. However, their sensitivity to low-concentration metabolites may be limited. This method is suitable for large hospitals or clinical centers that require rapid, high-throughput testing.
- **LC-MS/MS-based targeted metabolomics,** which offers high specificity and sensitivity and is well-suited for precise quantification of C5 and other key metabolites.
- **Plasma Metabolic Fingerprints (PMFs) Technology:** While promising in performance, PMFs technology is currently more suitable for research settings due to its high equipment cost.
- **Biosensor or microfluidic chip-based devices,** which could enable point-of-care testing with minimal sample volume and rapid turnaround time. While currently expensive, they show potential for early-stage exploration in research institutions or precision medicine centers.

2. Cost Considerations

As for cost estimates, we anticipate that, with optimization for high-throughput use and large-scale production, the per-test cost will be comparable to or lower than existing blood-based diagnostic assays. For example, enzymatic assays could bring costs below USD 20 per test, while LC-MS/MS-based assays might range from USD 30-50 per test depending on factors such as instrument throughput and reagent use. We are also working on optimizing assay designs and employing automation and mass production techniques to further reduce costs, ensuring broad accessibility in both resource-rich and resource-limited settings.

3. Next Steps

Moving forward, we plan to conduct a large-scale prospective cohort study across multiple tertiary hospitals in China. This study will establish diagnostic cutoffs, validate clinical performance across diverse populations, and meet regulatory requirements for in vitro diagnostic approval. These efforts are essential to ensure the clinical applicability and regulatory compliance of our metabolic-based gastric cancer detection approach.

We are confident that by addressing the clinical and economic aspects of commercialization, we can translate our research findings into a clinically deployable tool that supports early diagnosis, improves patient outcomes, and contributes to the advancement of precision oncology. We look forward to the potential impact of this work on clinical practice and public health.

Revisions: We have added a discussion on this topic in the Discussion section.

“To advance the clinical translation of our findings, we plan to conduct absolute quantitative metabolomics using a large-scale, multi-center patient cohort to establish reliable diagnostic thresholds for the key metabolites. In parallel, we are developing cost-effective detection platforms (e.g., ELISA, LC-MS/MS), with estimated per-test costs of USD 20–50, and will initiate a prospective validation study to assess diagnostic performance and facilitate regulatory approval. Based on the predictive model, commercial diagnostic kits and streamlined testing devices could be developed to enable early GC detection and guide clinical decision-making through individualized risk stratification. Furthermore, we aim to investigate the utility of plasma metabolite biomarkers in predicting GC risk and identifying high-risk individuals within asymptomatic populations, thereby supporting precision prevention and targeted intervention strategies.” (Page 8-9, Lines 301-312)

Comment 9: References: Update citations (e.g., "Chen et al, 2024b" is listed but absent in references).

Response: Thanks for your careful review. According to the citation format required by EMBO Molecular Medicine, all references have been formatted accordingly. The reference for "Chen et al., 2024b" is listed on **Page 20, Lines 763–766** of the reference section. We have double-checked the reference list to ensure that all in-text citations are correctly matched with their corresponding entries.

Chen Y-C, Malfertheiner P, Yu H-T, Kuo C-L, Chang Y-Y, Meng F-T, Wu Y-X, Hsiao J-L, Chen M-J, Lin K-P, et al (2024b) Global Prevalence of Helicobacter pylori Infection and Incidence of Gastric Cancer Between 1980 and 2022. *Gastroenterology* 166: 605–619

Comment 10: This study makes a substantial contribution to GC diagnostics and metabolic oncology. The diagnostic model holds immediate translational potential, while the mechanistic insights into C5 offer novel therapeutic targets. Addressing the above concerns will strengthen the manuscript for high-impact publication.

Response: We thank the reviewer for their thoughtful and constructive feedback. We agree that the integration of diagnostic modeling and mechanistic exploration of C5 represents a meaningful advance in the field of gastric cancer research. In response to the reviewer's constructive feedback, we have carefully revised the manuscript to address all concerns raised in previous comments. These revisions have been implemented to ensure the clarity, robustness, and translational relevance of our findings. We believe these improvements will further enhance the manuscript's impact and alignment with the journal's standards.

References (listed in alphabetical order)

Azuma T (2004) Helicobacter pylori CagA protein variation associated with gastric cancer in Asia. *Journal of Gastroenterology* 39: 97–103

Bertuccio P, Rosato V, Andreano A, Ferraroni M, Decarli A, Edefonti V & La Vecchia C (2013) Dietary patterns and gastric cancer risk: a systematic review and meta-analysis. *Ann Oncol* 24: 1450–1458

Chen Y, Wang B, Zhao Y, Shao X, Wang M, Ma F, Yang L, Nie M, Jin P, Yao K, et al (2024a) Metabolomic machine learning predictor for diagnosis and prognosis of gastric cancer. *Nat Commun* 15: 1657

Choi BH, Hyun S & Koo S-H (2024) The role of BCAA metabolism in metabolic health and disease. *Exp Mol Med* 56: 1552–1559

Demšar J, Curk T, Erjavec A, et al. (2013) Orange: data mining Toolbox in python. *J Mach Learn Res*: 2349–53

Dimou A, Tsimihodimos V & Bairaktari E (2022) The Critical Role of the Branched Chain Amino Acids (BCAAs) Catabolism-Regulating Enzymes, Branched-Chain Aminotransferase (BCAT) and Branched-Chain α -Keto Acid Dehydrogenase (BCKD), in Human Pathophysiology. *IJMS* 23: 4022

Fang X, Wei J, He X, An P, Wang H, Jiang L, Shao D, Liang H, Li Y, Wang F, et al (2015) Landscape of dietary factors associated with risk of gastric cancer: A systematic review and dose-response meta-analysis of prospective cohort studies. *European Journal of Cancer* 51: 2820–2832

Hooi JKY, Lai WY, Ng WK, Suen MMY, Underwood FE, Tanyingoh D, Malfertheiner P, Graham DY, Wong VWS, Wu JCY, et al (2017) Global Prevalence of *Helicobacter pylori* Infection: Systematic Review and Meta-Analysis. *Gastroenterology* 153: 420–429

Huang S, Guo Y, Li Z-W, Shui G, Tian H, Li B-W, Kadeerhan G, Li Z-X, Li X, Zhang Y, et al (2021) Identification and Validation of Plasma Metabolomic Signatures in Precancerous Gastric Lesions That Progress to Cancer. *JAMA Netw Open* 4: e2114186

Li P, Gao S, Qu W, Li Y & Liu Z (2024a) Chemo-Selective Single-Cell Metabolomics Reveals the Spatiotemporal Behavior of Exogenous Pollutants During *Xenopus Laevis* Embryogenesis. *Advanced Science* 11: 2305401

- Liu Q, Yu X, Jia F, Wen R, Sun C & Yu Q (2025) Comprehensive analyses of meat quality and metabolome alterations with aging under different aging methods in beef. *Food Chemistry* 472: 142936
- Pang Z, Lu Y, Zhou G, Hui F, Xu L, Viau C, Spigelman AF, MacDonald PE, Wishart DS, Li S, et al (2024) MetaboAnalyst 6.0: towards a unified platform for metabolomics data processing, analysis and interpretation. *Nucleic Acids Research* 52: W398–W406
- Park JY, Forman D, Waskito LA, Yamaoka Y & Crabtree JE (2018) Epidemiology of *Helicobacter pylori* and CagA-Positive Infections and Global Variations in Gastric Cancer. *Toxins (Basel)* 10: 163
- Qian L, Li N, Lu X-C, Xu M, Liu Y, Li K, Zhang Y, Hu K, Qi Y-T, Yao J, et al (2023) Enhanced BCAT1 activity and BCAA metabolism promotes RhoC activity in cancer progression. *Nat Metab* 5: 1159–1173
- Shu X, Zhan P-P, Sun L-X, Yu L, Liu J, Sun L-C, Yang Z-H, Ran Y-L & Sun Y-M (2021) BCAT1 Activates PI3K/AKT/mTOR Pathway and Contributes to the Angiogenesis and Tumorigenicity of Gastric Cancer. *Front Cell Dev Biol* 9: 659260
- Tourette E, Torres RC, Svensson SL, Matsumoto T, Miftahussurur M, Fauzia KA, Alfaray RI, Vilaichone R-K, Tuan VP, *Helicobacter Genomics Consortium*, et al (2024) An ancient ecospecies of *Helicobacter pylori*. *Nature* 635: 178–185
- Wei R & Wang J (2023) Left-Censored Missing Value Imputation Approach for MS-Based Proteomics Data with GSimp. In *Statistical Analysis of Proteomic Data*, Burger T (ed) pp 119–129. New York, NY: Springer US
- Wu B, Yang D, Yang S & Zhang G (2021) Dietary Salt Intake and Gastric Cancer Risk: A Systematic Review and Meta-Analysis. *Front Nutr* 8: 801228
- Xu Y, Yu W, Yang T, Zhang M, Liang C, Cai X & Shao Q (2018) Overexpression of BCAT1 is a prognostic marker in gastric cancer. *Human Pathology* 75: 41–46
- Xu Z, Huang Y, Hu C, Du L, Du Y-A, Zhang Y, Qin J, Liu W, Wang R, Yang S, et al (2023) Efficient plasma metabolic fingerprinting as a novel tool for diagnosis and prognosis of gastric cancer: a large-scale, multicentre study. *Gut* 72: 2051–2067

Yurekten O, Payne T, Tejera N, Amaladoss FX, Martin C, Williams M & O'Donovan C (2024) MetaboLights: open data repository for metabolomics. *Nucleic Acids Research* 52: D640–D646

Zhou X, Hou G, Wang X, Peng Z, Yin X, Yang J, Wang S, He Y, Wang Y, Sui J, et al (2024) Metabolomic studies reveal and validate potential biomarkers of diabetic retinopathy in two Chinese datasets with type 2 diabetes: a cross-sectional study. *Cardiovasc Diabetol* 23: 439

Zhu Y, Burg T, Neyrinck K, Vervliet T, Nami F, Vervoort E, Ahuja K, Sassano ML, Chai YC, Tharkeshwar AK, et al (2024) Disruption of MAM integrity in mutant FUS oligodendroglial progenitors from hiPSCs. *Acta Neuropathol* 147: 6

28th Aug 2025

Dear Prof. Du,

Thank you for the submission of your revised manuscript to EMBO Molecular Medicine. We have now heard back from the two referees who agreed to re-evaluate your manuscript. As you will see from their reports below, while referee #2 recommends publication of the revised manuscript, referee #1 remains critical regarding batch effects, MR validity, figure accuracy, and contextualization of prior studies. We concluded that raised concerns are justified and should be addressed in a final round of major revision.

Further consideration of a revision that addresses reviewer's concerns in full will entail an additional round of review. Acceptance or rejection of the manuscript will depend on the completeness of your responses included in the next, final version of the manuscript. For this reason, and to save you from any frustrations in the end, I would strongly advise against returning an incomplete revision.

In addition, for the next submission please provide institutional email address for co-corresponding author Yongjie Xu and correct name discrepancy Xiang-Dong Cheng vs Xiangdong Cheng in the manuscript and our submission system. In Appendix, please add page numbers to the table of contents and correct the nomenclature to 'Appendix Figure S1' etc. and 'Appendix Table S1' etc. Remove author contributions from the manuscript text. Also, please address all comments suggested by our data editors listed below:

- 1) Please note that the exact p values are not provided in the legends of figures 4A, 6A-C, F; 7A, E, F, G.
- 2) Please indicate the statistical test used for data analysis in the legends of figures 2B, 4C, 5A-C.
- 3) Please note that the box plots need to be defined in terms of minima, maxima, centre, bounds of box and whiskers, and percentile in the legends of figures 4A, 6F, 8A-C.
- 4) Please note that information related to n is missing in the legends of figures 3F, H; 4A, D; 8A-C.
- 5) Please note that the error bars are not defined in the legends of figures 3F, H; 4D, 6A-C; 7A, E, F, G.
- 6) Please note that scale bar and its definition are missing for figure 7E.

We would welcome the submission of a revised version within three months for further consideration. Please let us know if you require longer to complete the revision.

I look forward to receiving your revised manuscript.

Yours sincerely,

Zeljko Durdevic

Zeljko Durdevic
Senior Editor
EMBO Molecular Medicine

We require:

- 1) A .docx formatted version of the manuscript text (including legends for main figures, EV figures and tables). Please make sure that the changes are highlighted to be clearly visible.
- 2) Individual production quality figure files as .eps, .tif, .jpg (one file per figure). For guidance, download the 'Figure Guide PDF':

(<https://www.embopress.org/page/journal/17574684/authorguide#figureformat>).

3) A .docx formatted letter INCLUDING the reviewers' reports and your detailed point-by-point responses to their comments. As part of the EMBO Press transparent editorial process, the point-by-point response is part of the Review Process File (RPF), which will be published alongside your paper.

4) A complete author checklist, which you can download from our author guidelines (<https://www.embopress.org/page/journal/17574684/authorguide#submissionofrevisions>). Please insert information in the checklist that is also reflected in the manuscript. The completed author checklist will also be part of the RPF.

6) It is mandatory to include a 'Data Availability' section after the Materials and Methods. Before submitting your revision, primary datasets produced in this study need to be deposited in an appropriate public database, and the accession numbers and database listed under 'Data Availability'. Please remember to provide a reviewer password if the datasets are not yet public (see <https://www.embopress.org/page/journal/17574684/authorguide#dataavailability>).

.

12) Author contributions: You will be asked to provide CRediT (Contributor Role Taxonomy) terms in the submission system. These replace a narrative author contribution section in the manuscript.

13) A Conflict of Interest statement should be provided in the main text.

14) Every published paper now includes a 'Synopsis' to further enhance discoverability. Synopses are displayed on the journal webpage and are freely accessible to all readers. They include a short stand first (maximum of 300 characters, including space) as well as 2-5 one-sentences bullet points that summarizes the paper. Please write the bullet points to summarize the key NEW findings. They should be designed to be complementary to the abstract - i.e. not repeat the same text. We encourage inclusion of key acronyms and quantitative information (maximum of 30 words / bullet point). Please use the passive voice. Please attach these in a separate file or send them by email, we will incorporate them accordingly.

15) Include a Reagents and Tools Table as part of the Methods section, which can be downloaded from our author guidelines (<https://www.embopress.org/page/journal/17574684/authorguide#structuredmethods>)

***** Reviewer's comments *****

Referee #1 (Remarks for Author):

The revised manuscript is substantially improved, with clearer structure, enhanced methodological transparency, and strengthened biological validation. In particular, the comparative evaluation with recent diagnostic models, detailed reporting of classifier parameters with open-source workflows, and targeted validation of C5 notably increase the rigor and reproducibility of the study.

However, several critical issues remain:

1. Batch effects: The use of datasets from different institutions and LC-MS platforms raises concerns of confounding by technical variability. The manuscript does not report any batch correction or harmonization, making claims of platform-independent reproducibility insufficiently supported.
 2. Cross-population MR analysis: Applying European-derived genetic instruments to a Chinese cohort introduces population-specific biases (allele frequency, LD structure, metabolite-gene associations). Without sensitivity analyses or justification, the validity of causal inferences remains uncertain.
 3. Figure accuracy: Mislabeling in Figure 4D highlights insufficient quality control. All figures should be carefully re-verified to ensure correct and consistent labeling.
 4. Contextualization of prior studies: On Page 19, Line 703, the authors state that "these studies often focused on small cohorts or single-center datasets, limiting the generalizability of their findings." This characterization is inaccurate in the case of Chen et al. (Nat Commun, 2024), which was based on a large, multi-center cohort. Such misrepresentation weakens the contextualization of the current study within the existing literature. The statement should be revised to accurately reflect prior work, thereby ensuring scientific integrity and appropriate acknowledgment of the scope and impact of related studies.
- In summary, while the manuscript is stronger and more transparent, unresolved concerns regarding batch effects, MR validity, figure accuracy, and contextualization of prior studies must be addressed before the conclusions can be considered fully robust and generalizable.

Referee #2 (Comments on Novelty/Model System for Author):

The author solved the previous problem well, and the revised manuscript is now suitable for publication.

Referee #1: The revised manuscript is substantially improved, with clearer structure, enhanced methodological transparency, and strengthened biological validation. In particular, the comparative evaluation with recent diagnostic models, detailed reporting of classifier parameters with open-source workflows, and targeted validation of C5 notably increase the rigor and reproducibility of the study.

However, several critical issues remain:

1. Batch effects: The use of datasets from different institutions and LC-MS platforms raises concerns of confounding by technical variability. The manuscript does not report any batch correction or harmonization, making claims of platform-independent reproducibility insufficiently supported.

Response: We sincerely thank the reviewer for raising this important concern. The reviewer is correct that technical differences may exist across different LC-MS platforms, which is indeed a critical consideration in metabolomics research.

In our study:

- The discovery cohort was analyzed in a single batch on Platform A (Calibra).
- The validation cohort was analyzed in a single batch on Platform B (Shanghai Jiao Tong University).

Thus, no conventional intra-platform “batch effects” (i.e., multi-batch variability within the same instrument) were present. The key issue here is cross-platform variability. Importantly, our study was intentionally designed as a cross-platform validation. Validation on the same platform may still be influenced by platform-specific artifacts, whereas testing across two independent LC-MS systems provides stronger evidence of reproducibility.

To address potential cross-platform variability, we implemented the following measures:

1. Data preprocessing: Within each cohort, raw metabolite intensities were normalized by the sum of peak areas, log₂-transformed, and auto-scaled (mean-centering and division by each variable’s standard deviation) to reduce intra-cohort variability.
2. Cross-platform validation: We did not merge datasets from different platforms. Instead, we identified differential metabolites in the discovery set and independently validated them in the external cohort. The metabolite panel demonstrated consistent predictive performance across platforms, underscoring its robustness.

3. Manuscript revisions: We have revised the Methods to clarify that both cohorts were analyzed within a single batch, to emphasize that our study demonstrates reproducibility across platforms rather than correction of multi-batch technical variability.

Revisions: “The discovery cohort was analyzed in a single batch using an LC-MS platform at Calibra, and the external validation cohort was analyzed in a single batch using an independent LC-MS platform at Shanghai Jiao Tong University. Within each cohort, raw metabolite intensities were normalized by the sum of peak areas, log₂-transformed, and auto-scaled (mean-centering and division by each variable’s standard deviation) to reduce intra-cohort variability. No datasets from different platforms were merged. Instead, differential metabolites were identified in the discovery set and subsequently tested in the independent validation set to evaluate cross-platform reproducibility.” (Page 17, Lines 650-658, Methods section).

As our results show, the significant biological trends observed in the discovery cohort were successfully reproduced in an independent validation cohort measured on a different LC-MS platform. This strongly supports that our findings represent genuine biological signals rather than platform-specific artifacts.

We therefore believe that this cross-platform validation strategy is not a limitation, but a major strength of our study, providing stronger generalizability than same-platform multi-batch corrections.

2. Cross-population MR analysis: Applying European-derived genetic instruments to a Chinese cohort introduces population-specific biases (allele frequency, LD structure, metabolite-gene associations). Without sensitivity analyses or justification, the validity of causal inferences remains uncertain.

Response: We thank the reviewer for highlighting this important point regarding potential population-specific biases in Mendelian Randomization (MR) analyses. Large-scale, publicly available GWAS datasets for metabolites in Chinese or East Asian populations remain limited. In contrast, well-powered European GWAS offer broader metabolite coverage and stronger instrument strength. To our knowledge, the largest published genetic mapping of metabolome study in an East Asian population was reported by Cheng et al. (2025) [PMID: 39837327]; however, aggregated summary statistics are not publicly accessible for secondary analyses. Accordingly, previous MR

studies have also relied on European-derived instruments when ancestry-specific data were unavailable, allowing preliminary causal inference while acknowledging associated limitations [PMID: 38648753, 40157505].

To further evaluate the robustness of our results, we conducted multiple standard MR sensitivity analyses (Maximum likelihood, Simple median, Weighted median, MR-Egger), which are designed to be more resistant to potential violations such as weak or invalid instruments and horizontal pleiotropy. Notably, the Maximum likelihood method produced results that were consistent with those of the primary IVW analysis (Fig 4E).

In addition, to directly address potential ancestry-related concerns raised by the reviewer, we conducted a focused literature search on isovalerylcarnitine (C5), one of our key metabolites of interest. We identified a GWAS performed in an Asian population and carried out a two-sample MR analysis within the Biobank Japan gastric cancer cohort [PMID: 34594039; 40360795]. Although the association of isovalerylcarnitine (C5) did not reach statistical significance in this Asian cohort, the effect estimates were consistent in direction with those observed in our primary European-derived MR analysis. The limited number of instrumental variables and differences in allele frequencies and linkage disequilibrium structures across populations may have reduced statistical power.

To further consolidate the evidence, we performed a meta-analysis combining the Asian cohort results with our primary findings, which yielded a statistically significant association (OR = 0.83, 95% CI: 0.70–0.97). These results provide additional support for the robustness of our conclusions and suggest that the observed effect of isovalerylcarnitine (C5) on GC risk is likely generalizable across populations (**Table R1**).

Table R1 Mendelian Randomization Estimates of Isovalerylcarnitine (C5) on Gastric Cancer Risk: Evidence from Our Study, Asian Cohorts, and Meta-analysis

	OR	95% CI	P	PMID
Our study	0.833	0.697-0.995	0.044	36635386
Asian	0.794	0.525-1.199	0.079	34594039; 40360795
Meta (Fixed-effect Model)	0.827	0.702-0.974	0.023	

Importantly, the functional role of isovalerylcarnitine (C5) was further validated in biological experiments, where we demonstrated that isovalerylcarnitine (C5) inhibits GC progression. This experimental validation strengthens the causal inference suggested by

the MR analyses and provides mechanistic support for its potential tumor-suppressive role in gastric cancer.

Finally, we have explicitly acknowledged this limitation in the revised manuscript and highlighted the need for future large-scale metabolomics GWAS in Chinese or East Asian populations to validate and extend our findings. We believe these clarifications reinforce the robustness of our conclusions and their relevance across ancestries.

Revision: “Finally, our MR analysis relied primarily on European-derived genetic instruments due to the scarcity of large-scale metabolomics GWAS in East Asian populations. Although supplementary analyses in Asian cohorts (e.g., Biobank Japan) supported the robustness of our findings, future large-scale GWAS in Chinese or East Asian populations are warranted to fully validate causal inferences and ensure the generalizability of our results across ancestries.” (Page 11, Lines 416-421, Discussion section).

3. Figure accuracy: Mislabeling in Figure 4D highlights insufficient quality control. All figures should be carefully re-verified to ensure correct and consistent labeling.

Response: We thank the reviewer for pointing out the labeling issue in Figure 4D. We sincerely apologize for this oversight. The y-axis label in Figure 4D has now been corrected to “SNP effect on gastric cancer”. In addition, we have carefully re-checked all figures throughout the manuscript to ensure that labels, legends, and axes are accurate and consistent. We believe these revisions address the reviewer’s concerns regarding figure quality and clarity.

4. Contextualization of prior studies: On Page 19, Line 703, the authors state that “these studies often focused on small cohorts or single-center datasets, limiting the generalizability of their findings.” This characterization is inaccurate in the case of Chen et al. (Nat Commun, 2024), which was based on a large, multi-center cohort. Such misrepresentation weakens the contextualization of the current study within the existing literature. The statement should be revised to accurately reflect prior work, thereby ensuring scientific integrity and appropriate acknowledgment of the scope and impact of related studies.

Response: We thank the reviewer for pointing out this important issue. We have revised the relevant text on Page 19, Line 703, to accurately acknowledge prior work.

Revision: “Notably, a recent large multi-center study by *Chen et al.* (Nat Commun, 2024) demonstrated the diagnostic potential of metabolomics in GC using an external cohort. However, many earlier studies were limited by small sample sizes or single-center designs, which may constrain the generalizability of their findings. Moreover, although certain metabolic biomarkers have been implicated in GC risk, their functional roles in tumor progression and the underlying molecular mechanisms remain poorly understood. This knowledge gap has hindered the translation of metabolic discoveries into clinically actionable biomarkers or therapeutic targets for GC.” (**Page 19, Lines 695-698**).

This revision ensures accurate contextualization of prior studies, appropriately recognizes the scope and impact of *Chen et al.*'s work, and maintains scientific integrity in situating our study within the existing literature.

In summary, while the manuscript is stronger and more transparent, unresolved concerns regarding batch effects, MR validity, figure accuracy, and contextualization of prior studies must be addressed before the conclusions can be considered fully robust and generalizable.

Response: We have carefully addressed all reviewer concerns regarding technical reproducibility, MR analyses, figure accuracy, and literature contextualization. We believe these revisions substantially enhance the robustness, transparency, and generalizability of our conclusions, and we hope the reviewer finds the manuscript now suitable for publication.

We are deeply grateful to the reviewer for their insightful comments, which have not only strengthened the scientific rigor of this study but have also provided valuable guidance for our ongoing investigations in this field. The constructive critique has significantly enhanced the quality and impact of our work.

Referee #2 (Comments on Novelty/Model System for Author):

The author solved the previous problem well, and the revised manuscript is now suitable for publication.

Response: We sincerely thank the reviewer for the positive evaluation and for acknowledging our revisions. We are delighted that the manuscript is now considered suitable for publication. Your constructive comments have greatly improved the clarity and rigor of our work.

29th Sep 2025

Dear Prof. Du,

Thank you for the submission of your revised manuscript to EMBO Molecular Medicine. I am pleased to inform you that we will be able to accept your manuscript pending the following final amendments:

- 1) Please address referee's minor point.
- 2) Author checklist: Please submit a complete checklist as an excel file.
- 3) In the main manuscript file, please do the following:
 - In Methods, provide the antibody dilutions that were used for each antibody.
- 4) Synopsis:
 - Please check your synopsis text and image before submission with your revised manuscript. Please be aware that in the proof stage minor corrections only are allowed (e.g., typos).
- 5) As part of the EMBO Publications transparent editorial process initiative (see our Editorial at <http://embomolmed.embopress.org/content/2/9/329>), EMBO Molecular Medicine will publish online a Review Process File (RPF) to accompany accepted manuscripts. This file will be published in conjunction with your paper and will include the anonymous referee reports, your point-by-point response and all pertinent correspondence relating to the manuscript. Let us know whether you agree with the publication of the RPF and as here, if you want to remove or not any figures from it prior to publication. Please note that the Authors checklist will be published at the end of the RPF.
- 6) Please provide a point-by-point letter INCLUDING my comments as well as the reviewer's reports and your detailed responses (as Word file).

I look forward to reading a new revised version of your manuscript as soon as possible.

Yours sincerely,

Zeljko Durdevic

Zeljko Durdevic
Senior Editor
EMBO Molecular Medicine

*** Instructions to submit your revised manuscript ***

- 1) a .docx formatted version of the manuscript text (including Figure legends and tables)
- 2) Separate figure files*
- 3) supplemental information as Expanded View and/or Appendix. Please carefully check the authors guidelines for formatting Expanded view and Appendix figures and tables at <https://www.embopress.org/page/journal/17574684/authorguide#expandedview>

4) a letter INCLUDING the reviewer's reports and your detailed responses to their comments (as Word file).

5) The paper explained: EMBO Molecular Medicine articles are accompanied by a summary of the articles to emphasize the major findings in the paper and their medical implications for the non-specialist reader. Please provide a draft summary of your article highlighting

This may be edited to ensure that readers understand the significance and context of the research.

Please refer to any of our published articles for an example.

6) Author contributions: the contribution of every author must be detailed in a separate section.

7) EMBO Molecular Medicine now requires a complete author checklist

(<https://www.embopress.org/page/journal/17574684/authorguide>) to be submitted with all revised manuscripts. Please use the checklist as guideline for the sort of information we need WITHIN the manuscript. The checklist should only be filled with page numbers where the information can be found. This is particularly important for animal reporting, antibody dilutions (missing) and exact values and n that should be indicated instead of a range.

8) Every published paper now includes a 'Synopsis' to further enhance discoverability. Synopses are displayed on the journal webpage and are freely accessible to all readers. They include a short stand first (maximum of 300 characters, including space) as well as 2-5 one sentence bullet points that summarise the paper. Please write the bullet points to summarise the key NEW findings. They should be designed to be complementary to the abstract - i.e. not repeat the same text. We encourage inclusion of key acronyms and quantitative information (maximum of 30 words / bullet point). Please use the passive voice. Please attach these in a separate file or send them by email, we will incorporate them accordingly.

You are also welcome to suggest a striking image or visual abstract to illustrate your article. If you do please provide a jpeg file 550 px-wide x 300-600px high.

9) A Conflict of Interest statement should be provided in the main text

10) Please note that we now mandate that all corresponding authors list an ORCID digital identifier. This takes <90 seconds to complete. We encourage all authors to supply an ORCID identifier, which will be linked to their name for unambiguous name identification.

Currently, our records indicate that the ORCID for your account is 0000-0002-1992-7784.

Link Not Available

11) Include a Reagents and Tools Table as part of the Methods section, which can be downloaded from our author guidelines (<https://www.embopress.org/page/journal/17574684/authorguide#structuredmethods>)

Photos 400-800 DPI

*Additional important information regarding figures and illustrations can be found at

<https://bit.ly/EMBOPressFigurePreparationGuideline>. See also figure legend preparation guidelines:

<https://www.embopress.org/page/journal/17574684/authorguide#figureformat>

***** Reviewer's comments *****

Referee #1 (Remarks for Author):

The manuscript has been substantially strengthened by the revisions. The additional sensitivity analyses reinforce the robustness of the causal inference, the corrected figure annotations improve accuracy and clarity, and the expanded discussion offers a stronger integration with prior literature.

The modeling strategy-training and feature selection within the discovery cohort and applying fixed parameters to the validation cohort-is sound and supports robustness. Nonetheless, as platform and cohort remain confounded, the conclusions should be framed in terms of cross-cohort generalizability rather than strict platform independence.

Overall, the revisions have satisfactorily addressed prior concerns, and the study is suitable for publication.

The authors addressed the remaining editorial issues.

9th Oct 2025

Dear Prof. Du,

We are pleased to inform you that your manuscript is accepted for publication and is now being sent to our publisher to be included in the next available issue of EMBO Molecular Medicine.

Zeljko Durdevic
Senior Editor
EMBO Molecular Medicine
